# ENHANCING META LEARNING VIA MULTI-OBJECTIVE SOFT IMPROVEMENT FUNCTIONS

**Runsheng Yu**[1]**, Weiyu Chen**[1]**, Xinrun Wang**[2]**, James T. Kwok**[1]
[1]Department of Computer Science and Engineering, Hong Kong University of Science and Technology
[2]School of Computer Science and Engineering, Nanyang Technological University
`runshengyu@gmail.com`, `wchenbx@cse.ust.hk`,
`xinrun.wang@ntu.edu.sg`, `jamesk@cse.ust.hk`

## ABSTRACT

Meta-learning tries to leverage information from similar learning tasks. In the commonly-used bilevel optimization formulation, the shared parameter is learned in the outer loop by minimizing the average loss over all tasks. However, the converged solution may be compromised in that it only focuses on optimizing on a small subset of tasks. To alleviate this problem, we consider meta-learning as a multi-objective optimization (MOO) problem, in which each task is an objective. However, existing MOO solvers need to access all the objectives' gradients in each iteration, and cannot scale to the huge number of tasks in typical meta-learning settings. To alleviate this problem, we propose a scalable gradient-based solver with the use of mini-batch. We provide theoretical guarantees on the Pareto optimality or Pareto stationarity of the converged solution. Empirical studies on various machine learning settings demonstrate that the proposed method is efficient, and achieves better performance than the baselines, particularly on improving the performance of the poorly-performing tasks and thus alleviating the compromising phenomenon.

## 1 INTRODUCTION

Meta-learning, also known as "learning to learn", aims to enable models to learn more effectively by leveraging information from many similar learning tasks (Hospedales et al., 2020). In recent years, meta-learning has received much attention for its fast adaptation to new learning scenarios with limited data (Kao et al., 2021; Finn et al., 2017; Snell et al., 2017; Lee et al., 2019; Nichol et al., 2018; Deleu et al., 2022; Rajeswaran et al., 2019; Vilalta & Drissi, 2002). It is usually formulated as a bi-level optimization problem (Franceschi et al., 2018; Hong et al., 2020), which finds task-specific parameters in the inner level and minimizes the average loss over tasks in the outer level.

Recently, Wang et al. (2021) reformulate meta-learning as a multi-task learning problem. From this perspective, minimizing the average loss in the outer level using (stochastic) gradient descent may not always be desirable. Specifically, it may suffer from the compromising (or conflicting) phenomenon, in which the converged solution only focuses on minimizing the losses of a small subset of tasks while ignoring the others (Yu et al., 2020; Liu et al., 2021a; Sener & Koltun, 2018). This compromised solution may thus lead to poor performance.

To alleviate this problem, we propose reformulating meta-learning as a multi-objective optimization (MOO) problem, in which each task is an objective. The performance of all tasks (objectives) are then considered during optimization (Emmerich & Deutz, 2018). A popular class of MOO solvers is the gradient-based approach (Liu et al., 2021a; Yu et al., 2020; Sener & Koltun, 2018; Navon et al., 2022; Liu et al., 2021b), with prominent examples such as the multiple-gradient descent algorithm (MGDA) (Désidéri, 2012; Sener & Koltun, 2018), PCGard (Yu et al., 2020), and CAGard (Liu et al., 2021a). In each iteration, they find a common descent direction among all objective gradients, instead of simply optimizing the average performance over all objectives.

Existing gradient-based MOO methods require using gradients from all the objectives. However, when formulating meta-learning as a MOO problem with each task being an objective, computing all these gradients in each iteration can become very expensive, as the number of objectives (i.e., tasks)

can be huge. For example, in 5-way 1-shot classification on the *miniImageNet* data, the total number of meta-training tasks is $\binom{64}{5} \approx 7 \times 10^6$.

To address this challenge, we propose a scalable MOO solver by using the improvement function (Miettinen & Mäkelä, 1995; Mäkelä et al., 2016; Montonen et al., 2018) with the help of mini-batch. On the other hand, we show that a trivial extension of existing gradient-based MOO methods with the use of mini-batch does not guarantee Pareto optimality and has poor performance in practice.

Our main contributions are as follows: (i) To alleviate the compromising phenomenon, we reformulate meta-learning as a multi-objective optimization problem in which each task is an objective; (ii) To handle the possibly huge number of tasks, we propose a scalable gradient-based solver. (iii) We provide theoretical guarantees on the Pareto optimality or Pareto stationarity of the converged solution. (iv) Empirical studies on few-shot regression, few-shot classification, and reinforcement learning demonstrate that the proposed method achieves better performance, particularly in improving the performance of the poorly-performing tasks and thus alleviating the compromising phenomenon.

## 2 BACKGROUND

**Multi-Objective Optimization (MOO).** In MOO (Marler & Arora, 2004), one aims to minimize[1] $m \geq 2$ objectives $f_1(x), \ldots, f_m(x)$:

$$\min_x [f_1(x), \ldots, f_m(x)]. \tag{1}$$

**Definition 2.1.** *(Global Pareto optimality) (Miettinen, 2012; Mäkelä et al., 2016) $x^*$ is global Pareto optimal if there does not exist another $x$ such that $f_\tau(x^*) \geq f_\tau(x)$ for all $\tau \in \{1, \ldots, m\}$, and $f_{\tau'}(x^*) > f_{\tau'}(x)$ for at least one $\tau' \in \{1, \ldots, m\}$.*

The *Pareto front* (PF) is the set of multi-objective values of all global Pareto-optimal solutions.

**Definition 2.2.** *(Pareto stationarity) (Miettinen, 2012; Désidéri, 2012) $x^*$ is Pareto-stationary if there exist $\{u_\tau\}_{\tau=1}^m$ such that $\|\sum_{\tau=1}^m u_\tau \nabla_x f_\tau(x)\| = 0, u_\tau \geq 0 \,\forall \tau$ and $\sum_{\tau=1}^m u_\tau = 1$.*

Note that global Pareto optimal solutions are also Pareto stationary (Désidéri, 2012). Analogous to the extension from a stationary point to an $\epsilon$-stationary point (Lin et al., 2020), we extend Pareto stationarity to $\epsilon$-Pareto stationarity. Obviously, 0-Pareto stationarity reduces to Pareto stationarity.

**Definition 2.3.** *($\epsilon$-Pareto stationarity). For a given $\epsilon$, $x$ is $\epsilon$-Pareto-stationary iff there exist $\{u_\tau\}_{\tau=1}^m$ such that $\|\sum_{\tau=1}^m u_\tau \nabla_x f_\tau(x)\| \leq \epsilon, u_\tau \geq 0 \,\forall \tau$ and $\sum_{\tau=1}^m u_\tau = 1$.*

**Definition 2.4.** *(Improvement function) (Montonen et al., 2018) The improvement function of problem (1) is: $H(x, x') = \max_{\tau=1 \ldots, m} \{f_\tau(x) - f_\tau(x')\}$.*

Note that $x^*$ satisfying $x^* = \arg\min_x H(x, x^*)$ (intuitively, $x^*$ cannot be further improved) is Pareto stationary (Montonen et al., 2018). To find $x^*$, one can perform steepest descent on $H$:

$$x^{s+1} = x^s + \beta d^*, \quad d^* = \arg\min_d H(x^s + d, x^s) + \frac{\lambda'}{2}\|d\|^2, \tag{2}$$

where $x^s$ is the iterate at iteration $s$, $\beta$ is the learning rate satisfying $H(x^s + \beta d, x^s) < H(x^s, x^s)$, and $\lambda'$ is a hyper-parameter. It can be shown that when $s \to \infty$, $x^s$ is Pareto stationary (Montonen et al., 2018).

In this paper, we focus on gradient-based MOO methods, including MGDA (Désidéri, 2012; Sener & Koltun, 2018), PCGard (Yu et al., 2020), and CAGard (Liu et al., 2021a). They assign weights to each objective's gradient and find a common descent direction that decreases the losses of all objectives. For example, MGDA finds the direction $g^*(x) = \sum_{\tau=1}^m \gamma_\tau^* \nabla_x f_\tau(x)$ in each iteration, where

$$\{\gamma_\tau^*\} = \arg\min_{\{\gamma_\tau\}} \left\|\sum_{\tau=1}^m \gamma_\tau \nabla_x f_\tau(x)\right\|^2 \text{ s.t. } \sum_{\tau=1}^m \gamma_\tau = 1, \gamma_\tau \geq 0, \forall \tau. \tag{3}$$

**Meta-Learning.** Meta-learning aims to achieve good performance with limited data and computation (Hospedales et al., 2020). Most of them are gradient-based (Nichol et al., 2018; Deleu et al., 2022; Rajeswaran et al., 2019; Zhou et al., 2019; Shu et al., 2019) or metric-based (Snell et al., 2017;

---

[1]Without loss of generality, we consider minimization in this paper.

Lee et al., 2019; Vinyals et al., 2016). Let $\mathcal{T}$ be the set of all $m$ tasks, and $w$ be the shared model parameter. For a task $\tau \in \mathcal{T}$, let $D_\tau$ be its dataset and $\mathcal{L}_\tau$ the corresponding loss function. It tries to obtain task-specific parameter $w_\tau$ from the shared $w$ as $w_\tau^*(w)$. Meta-learning is usually formulated as the following bilevel optimization problem (Ji et al., 2021):

$$\min_w \sum_{\tau=1}^{m} \mathcal{L}_\tau(w_\tau) \text{ s.t. } w_\tau = w_\tau^*(w). \tag{4}$$

The inner subproblem learns the task-specific parameter $w_\tau$ for each $\tau$, while the outer subproblem learns $w$ by minimizing the average loss over tasks in $\mathcal{T}$. As $m$ can be very large, usually a mini-batch $B$ of tasks are uniformly sampled from $\mathcal{T}$, and $w$ is then updated as $w^{s+1} = w^s - \beta \frac{1}{|B|} \sum_{\tau \in B} \mathcal{L}_\tau(w_\tau^*(w))$ (Finn et al., 2017).

By taking each $\mathcal{L}_\tau(w_\tau^*(w))$ in (4) as an objective, this can be regarded as a weighted sum in multi-task learning (Wang et al., 2021). As observed in (Sener & Koltun, 2018; Yu et al., 2020), gradient descent on this weighted sum can suffer from the *compromising* phenomenon, in which the loss obtained on some task $\tau'$ can be much larger than the losses on the other tasks.

# 3  SOFT IMPROVEMENT MULTI-OBJECTIVE META-LEARNING (SIMOL)

We take the view of meta-learning as multi-task learning in (Wang et al., 2021) one step further and consider the meta-learning problem as the following multi-objective optimization (MOO) problem:

$$\min_w (\mathcal{L}_1(w_1^*(w)), \dots, \mathcal{L}_m(w_m^*(w))), \tag{5}$$

in which each task corresponds to an objective. This considers all the individual tasks instead of simply considering the total loss over all tasks (Liu et al., 2021a). Recently, Ye et al. (2021) also use multi-objective learning into meta-learning. However, their focus is not on addressing the compromising phenomenon and they do not treat each task as an objective. Instead, besides minimizing the average task loss in (4), they consider adding some other objectives such as robustness to adversarial attacks. Moreover, MGDA is still used to find the Pareto optimal solution. However, as in other gradient-based MOO methods (Yu et al., 2020; Liu et al., 2021a), MGDA requires collecting gradients from all $m$ objectives in each iteration (as can be seen from its optimization problem (3)). This is computationally feasible only when there are a small number of objectives.[2] When each task is treated as an objective, the number of objectives can easily be in the millions (as in performing 5-way 1-shot classification on *miniImageNet*).

Another widely adopted MOO based methods are the Chebyshev methods (Miettinen, 2012; Mao et al., 2020; Momma et al., 2022), which leverage the weighted Chebyshev problem to find the Pareto front. However, these methods also cannot handle a huge number of tasks, as the computational complexity per epoch for these methods is $O(m^2)$, where $m$ is the number of tasks.

To alleviate this problem, one solution is to use only a mini-batch of objectives in each iteration. For example, when a subset $B$ of objectives is used, MGDA's optimization problem in (3) becomes:

$$\min_{\{\gamma_\tau\}} \left\| \sum_{\tau \in B} \gamma_\tau \nabla_\tau(\mathcal{L}_\tau(w_\tau^*(w)) \right\|^2 \text{ s.t. } \sum_{\tau \in B} \gamma_\tau = 1, \gamma_\tau \geq 0, \forall \tau.$$

However, the descent direction then only considers objectives in $B$, and the original normalization constraint $\sum_{\tau=1}^{m} \gamma_\tau = 1$ in (3) is also changed to $\sum_{\tau \in B} \gamma_\tau = 1$. The obtained solution may no longer Pareto optimal. In the following, we demonstrate this by using a simple toy example with two objectives ($f_1(x)$ and $f_2(x)$, where $x \in \mathbb{R}^2$) from (Liu et al., 2021a; Navon et al., 2022).[3] As can be seen from Figure 1, mini-batch MGDA (with a mini-batch of 1) cannot converge to the Pareto front.

## 3.1  SOFT IMPROVEMENT FUNCTION

In this section, we propose a scalable MOO solver with the use of the improvement function (Montonen et al., 2018). The proposed solver is agnostic to the number of tasks, while still theoretically guaranteeing that the solution is Pareto optimal.

Using Definition 2.4, the improvement function for problem (5) is:

$$H(w, w') = \max_{\tau=1\dots,m} \left\{ \mathcal{L}_\tau(w_\tau^*(w)) - \mathcal{L}_\tau(w_\tau^*(w')) \right\}. \tag{6}$$

---

[2]In the meta-learning experiments of (Ye et al., 2021), they only consider two objectives.
[3]Definitions for $f_1$, $f_2$ and the environment setup are in Appendix A.

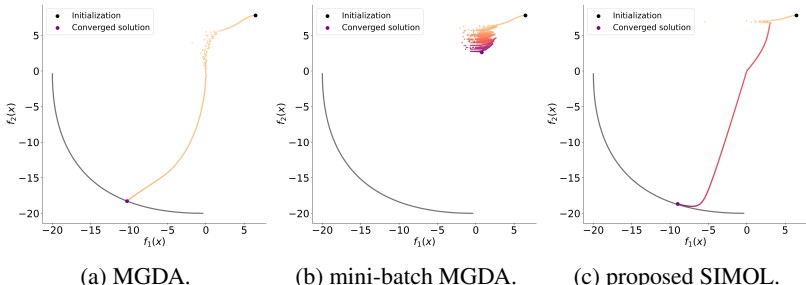

(a) MGDA.          (b) mini-batch MGDA.          (c) proposed SIMOL.

Figure 1: Convergence on a two-objective toy dataset with mini-batch size 1. The Pareto front is shown in black.

Consider the optimization problem

$$\max_\pi \mathbb{E}_{\tau \sim \pi} \left[ \mathcal{L}_\tau(w_\tau^*(w)) - \mathcal{L}_\tau(w_\tau^*(w')) \right], \tag{7}$$

where $\pi$ is a probability density function on $\tau$. The following Lemma shows that (6) and (7) are equivalent when $\pi$ is the Dirac delta distribution concentrated on the task corresponding to the maximum in (6). All the proofs are in Appendix C.

**Lemma 3.1.** $H(w, w') = \max_\pi \mathbb{E}_{\tau \sim \pi} \left[ \mathcal{L}_\tau(w_\tau^*(w)) - \mathcal{L}_\tau(w_\tau^*(w')) \right].$

Using (2) and Lemma 3.1, $w$ can be updated as

$$w^{s+1} = w^s + \beta d^*, \tag{8}$$

$$d^* = \arg\min_d \left( \max_\pi \mathbb{E}_{\tau \sim \pi} \left[ \mathcal{L}_\tau(w_\tau^*(w^s + d)) - \mathcal{L}_\tau(w_\tau^*(w^s)) \right] \right) + \frac{\lambda'}{2} \|d\|^2. \tag{9}$$

Taking the first-order approximation

$$\mathcal{L}_\tau(w_\tau^*(w^s + d)) \simeq \mathcal{L}_\tau(w_\tau^*(w^s)) + \nabla_w \mathcal{L}_\tau(w_\tau^*(w^s))^\top d, \tag{10}$$

the minimax theorem (Simons, 1995) can be used to swap the $\min$ and $\max$ operators in (9), as

$$(\pi^*, d^*) = \arg\max_\pi \left( \min_d \mathbb{E}_{\tau \sim \pi} \left[ \mathcal{L}_\tau(w_\tau^*(w^s + d)) - \mathcal{L}_\tau(w_\tau^*(w^s)) \right] + \frac{\lambda'}{2} \|d\|^2 \right). \tag{11}$$

The following Proposition shows that the inner minimization problem has a closed-form solution.

**Proposition 3.2.** $\min_d \mathbb{E}_{\tau \sim \pi} \left[ \mathcal{L}_\tau(w_\tau^*(w^s + d)) - \mathcal{L}_\tau(w_\tau^*(w^s)) \right] + \frac{\lambda'}{2} \|d\|^2 = \frac{-1}{2\lambda'} \| \mathbb{E}_{\tau \sim \pi} \nabla_w \mathcal{L}_\tau(w_\tau^*(w))|_{w=w^s} \|^2$, and the optimal $d$ is $d^* = -\frac{1}{\lambda'} \mathbb{E}_{\tau \sim \pi} [\nabla_w \mathcal{L}_\tau(w_\tau^*(w))|_{w=w^s}].$

The expectation in Proposition 3.2 requires sampling tasks from $\pi$. An easier alternative is to sample tasks from the uniform distribution $U(\cdot)$ over the set $\mathcal{T}$ of all tasks, and then weighting each sampled task $\tau$ with $r(\tau) \equiv \pi(\tau)/U(\tau)$. Note that

$$\mathbb{E}_{\tau \sim U} r(\tau) = \sum_\tau U(\tau) \pi(\tau)/U(\tau) = \sum_\tau \pi(\tau) = 1. \tag{12}$$

We further parameterize $r$ as a neural network $r_\theta$ with parameter $\theta$. Using Proposition 3.2, we can then rewrite (11) as

$$\theta^* = \arg\max_\theta \frac{-1}{2\lambda'} \| \mathbb{E}_{\tau \sim U} r_\theta(\tau) \nabla_w \mathcal{L}_\tau(w_\tau^*(w))|_{w=w^s} \|^2 - \frac{\lambda''}{2} (\mathbb{E}_{\tau \sim U} r_\theta(\tau) - 1)^2, \tag{13}$$

where the last term (with another hyper-parameter $\lambda''$) is a penalty for enforcing the constraint in (12). For notational simplicity, we denote the objective in (13) by $K(\theta)$.

In principle, $\theta^*$ can be obtained from (13) by gradient ascent. However, problem (13) involves an expectation over tasks. Recall that we have a total of $m$ tasks, and $m$ can be huge. Hence, using all of them to compute this expectation may not be feasible. Instead, Let $B$ be a mini-batch of $k$ tasks, and denote the the mini-batched version of the objective in (13) as:

$$\tilde{K}_B(\theta) \equiv \frac{-1}{2\hat{\lambda}'} \left\| \frac{1}{|B|} \sum_{\tau \in B} r_\theta(\tau) \nabla_w \mathcal{L}_\tau(w_\tau^*(w))|_{w=w^s} \right\|^2 - \frac{\hat{\lambda}''}{2} \left( \frac{1}{|B|} \sum_{\tau \in B} r_\theta(\tau) - 1 \right)^2,$$

where $\hat{\lambda}'$, $\hat{\lambda}''$ are another set of hyper-parameters (which will be set in Proposition 3.3) corresponding to $\lambda'$, $\lambda''$ in (13). Note that $\tilde{K}_\mathcal{T}(\theta) = K(\theta)$. Let $\mathcal{B}$ be the set of all size-$k$ mini-batches (with $k > 1$). The following Proposition bounds the difference between $K(\theta)$, the original objective in (13), and the version $\frac{1}{|\mathcal{B}|} \sum_{B \in \mathcal{B}} \tilde{K}_B(\theta)$ based on mini-batches.

**Proposition 3.3.** *Set* $\hat{\lambda}' = \frac{\lambda'}{C_1 |\mathcal{B}|}$ *and* $\hat{\lambda}'' = \lambda'' C_1 |\mathcal{B}| k^2$. *We have:*
$$\left(K(\theta) - \frac{1}{|\mathcal{B}|} \sum_{B \in \mathcal{B}} \tilde{K}_B(\theta)\right)^2 \leq \frac{C_2 k |\mathcal{B}| G_1}{\lambda'} + C_2 k |\mathcal{B}| \lambda'' G_2, \quad \text{where} \quad C_1 \equiv$$
$$\frac{k^2}{\binom{m-2}{k-2} m^2}, \quad \text{and} \quad C_2 \equiv \left[\binom{m-1}{k-1} - \frac{1}{2}\binom{m-2}{k-2}\right] / \binom{m-1}{k-1} m^2 \binom{m-2}{k-2}, G_1 \equiv$$
$$\frac{1}{k|\mathcal{B}|} \sum_{B \in \mathcal{B}} \sum_{\tau \in B} \|r_\theta(\tau) \nabla_w \mathcal{L}_\tau(w_\tau^*(w)|_{w=w^s})\|^2, \text{ and } G_2 \equiv \frac{1}{k|\mathcal{B}|} \sum_{B \in \mathcal{B}} \sum_{\tau \in B} [r_\theta(\tau) - 1]^2.$$

**Corollary 3.3.1.** *When* $k \ll m$, $\left(K(\theta) - \frac{1}{|\mathcal{B}|} \sum_{B \in \mathcal{B}} \tilde{K}_B(\theta)\right)^2 \leq \frac{G_1}{k\lambda'} + \frac{G_2}{k}\lambda''.$

In the experiments, $m \geq 10^6$, $k \approx 10^2$, and $G_1, G_2 \leq 10^4$. When $\lambda' \geq 10^2, \lambda'' \leq 10^{-2}$, we have $\frac{1}{|\mathcal{B}|} \sum_{B \in \mathcal{B}} \tilde{K}_B(\theta) \in [-10^3, -1]$ during training, and $(K(\theta) - \frac{1}{|\mathcal{B}|} \sum_{B \in \mathcal{B}} \tilde{K}_B(\theta))^2 \leq 0.2$ is small. Thus, Proposition 3.3 shows that $K(\theta)$ can be decomposed into mini-batches as $\frac{1}{|\mathcal{B}|} \sum_{B \in \mathcal{B}} \tilde{K}_B(\theta)$. This allows us to update $\theta$ by SGD over the task mini-batches as:

$$\theta^{s+1} = \theta^s + \beta' \nabla_\theta \tilde{K}_B(\theta^s), \tag{14}$$

where $\beta'$ is the learning rate. Similarly, we approximate $d^*$ by its mini-batch approximation $\tilde{d}^* = -\frac{1}{\lambda'|B|}[\sum_{\tau \in B} \nabla_w \mathcal{L}_\tau(w_\tau^*(w))|_{w=w^s}]$ and update $w$ as $w^{s+1} = w^s + \beta \tilde{d}^*$.

The whole procedure, which will be called Soft Improvement Multi-Objective Meta Learning (SIMOL), is shown in Algorithm 1. Step 4 trains the base learner. In the experiments, we use two popular meta-learning algorithms: MAML (Finn et al., 2017) and prototypical network (PN) (Snell et al., 2017). For MAML, the base learner is updated as

$$w_\tau^*(w) = w - \alpha \nabla_w \mathcal{L}_\tau(w). \tag{15}$$

For the PN, $w_\tau^*(w) = \frac{1}{|Q_\tau| N_C} \sum_{x \in Q_\tau} \frac{\exp(-\|f_w(x) - c_k\|^2)}{\sum_{k'} \exp(-\|f_w(x) - c_{k'}\|^2)}$, where $Q_\tau$ is the set of query examples for task $\tau$, $N_C$ is the number of classes per epoch, $f_w$ is the model with parameter $w$, $c_k = \frac{1}{|S_k|} \sum_{(x_i, y_i) \in S_k} f_w(x_i)$, and $S_k$ is the set of examples belonging to class $k$. Pseudo-codes for SIMOL-based MAML and PN are shown in Algorithms 2 and 3 of Appendix B, respectively.

---

**Algorithm 1:** Soft Improvement Multi-Objective Meta Learning (SIMOL)

---

**Input:** $\mathcal{T}$, batch size $k$, learning rates $\beta$ and $\beta'$ for $w$, $\tilde{d}^* = 0$ and $\theta$, respectively.

1 **for** $s = 1, 2, \ldots, S$ **do**
2 $\quad$ Reset $\tilde{d}^* = 0$;
3 $\quad$ **for** $\tau = 1, 2, \ldots, k$ **do**
4 $\quad\quad$ obtain $r_{\theta^s}(\tau) \nabla_{w^s} \mathcal{L}_\tau(w_\tau^*(w^s))$ for task $\tau$;
5 $\quad\quad$ $\tilde{d}^* = \tilde{d}^* - r_{\theta^s}(\tau) \nabla_{w^s} \mathcal{L}_\tau(w_\tau^*(w^s))$;
6 $\quad$ $w^{s+1} = w^s + \beta \frac{1}{k} \tilde{d}^*$;
7 $\quad$ $\theta^{s+1} = \theta^s + \beta' \frac{-1}{2\hat{\lambda}'} \nabla_{\theta^s} \left\| \frac{1}{k} \sum_{\tau \in B} r_{\theta^s}(\tau) \nabla_w \mathcal{L}_\tau(w_\tau^*(w))|_{w=w^{s+1}} \right\|^2 -$
$\quad\quad \nabla_{\theta^s} \frac{\hat{\lambda}''}{2} \left(\frac{1}{k} \sum_{\tau \in B} r_{\theta^s}(\tau) - 1\right)^2$;

---

## 4 CONVERGENCE ANALYSIS

**Lemma 4.1.** *Define*

$$R(\theta, w) \equiv \mathbb{E}_{\tau \sim U}\left[\perp (r_\theta(\tau))[\mathcal{L}_\tau(w_\tau^*(w))]\right] + \Delta(\theta) - \frac{\lambda''}{2}(\mathbb{E}_{\tau \sim U} r_\theta(\tau) - 1)^2, \tag{16}$$

where $\Delta(\theta) \equiv -\frac{1}{\lambda'}\mathbb{E}_{\tau\sim U} r_\theta(\tau) \perp [(\mathcal{L}_\tau(w_\tau^*(w) + d^*) - \mathcal{L}_\tau(w_\tau^*(w)))] \cdot \perp [\mathbb{E}_{\tau\sim U} r_\theta(\tau)[\mathcal{L}_\tau(w_\tau^*(w) + d^*) - \mathcal{L}_\tau(w_\tau^*(w))]]$, and $\perp$ is the stop gradient operator.[4] Then, $\nabla_w R(\theta, w)|_{w=w^s} = -d^*$, and $\nabla_\theta R(\theta, w^s) = \nabla_\theta K(\theta)$.

This allows interpreting the updates in (14) and (8) as performing Gradient Descent Ascent (GDA) (Singh et al., 2000) on (16). Thus, we can leverage game theoretical tools (Lin et al., 2020) in the analysis.

Let $\tilde{R}(\theta, w; B) \equiv \perp \frac{1}{|B|}\sum_{\tau\in B}[r_\theta(\tau)][\mathcal{L}_\tau(w_\tau^*(w)] - \frac{1}{|B|\lambda'}\sum_{\tau\in B} r_\theta(\tau)[\perp [(\mathcal{L}_\tau(w_\tau^*(w) + d^*) - \mathcal{L}_\tau(w_\tau^*(w))]\cdot \perp \left[\frac{1}{|B|}\sum_{\tau\in B} r_\theta(\tau)[\mathcal{L}_\tau(w_\tau^*(w) + d^*) - \mathcal{L}_\tau(w_\tau^*(w))]\right] - \frac{\lambda''}{2}(\frac{1}{|B|}\sum_{\tau\in B} r_\theta(\tau) - 1)^2$ be the mini-batch version of $R$, and $U(\mathcal{B})$ be the uniform distribution over task mini-batches. The following Theorem shows that Algorithm 1 converges to an $\epsilon$-Pareto stationary point of (5).

**Theorem 4.2.** *Assume that (i) $\mathcal{L}_\tau(w_\tau^*(\cdot))$ is L-smooth and $r_\theta(\cdot)$ is $\mu$-strongly concave. (ii) The domain of $\theta$ is a convex and bounded set with diameter $D > 0$, (iii) $\mathbb{E}_{B\sim U(\mathcal{B})}[\nabla_\theta \tilde{R}(\theta, w; B) - \nabla_\theta R(\theta, w)] = 0$, and $\mathbb{E}_{B\sim U(\mathcal{B})}\|\nabla_\theta \tilde{R}(\theta, w, B) - \nabla_\theta R(\theta, w)\|^2 \leq \sigma^2$. Assume the first-order approximation in (10), and take $\beta = \Theta\left(1/D^2\sigma^2(L^2 + \sigma^2)\right)$, $\beta' = \Theta(1/L\sigma^2)$. Algorithm 1 converges to an $\epsilon$-Pareto stationary point of (5) with a rate of $O(1/\epsilon^8)$. If $\mathcal{L}_\tau(w_\tau^*(w))$ is also $\mu'$-convex w.r.t. $w$ and $\epsilon = 0$, the 0-Pareto stationary point is also global Pareto optimal.*

Assumption (i) is commonly used in the literature (Collins et al., 2020; Zhou et al., 2021; Finn et al., 2019); while (ii) and (iii) are from (Lin et al., 2020). Theorem 4.2 shows that the proposed method can obtain an $\epsilon$-Pareto stationary point (or global Pareto optimal point for convex objectives) regardless of $m$, the number of tasks/objectives.

**Corollary 4.2.1.** *Consider the MAML base learner update in (15). Assume that $\nabla_w \mathcal{L}_\tau(w)$ is Hessian-Lipschitz continuous, bounded, Lipschitz-continuous, and $w$ is bounded. Then, Algorithm 1 converges to an $\epsilon$-Pareto stationary point of (5) with a rate of $O(1/\epsilon^8)$.*

Corollary 4.2.1 is an application of Theorem 4.2 revealing that SIMOL with MAML can also converge to a Pareto point. Convergence of the outer loop is slower than the $O(1/\epsilon^2)$ rate of standard MAML (Fallah et al., 2020). However, standard MAML only guarantees convergence to stationary points of $w$ but not to Pareto-stationary points. Moreover, as will be seen in Section 5.1, empirically, the proposed method has comparable or even slightly faster convergence speed than MAML and other meta learning baselines. Besides, most the gradient-based MOO approaches (except CAGrad (Liu et al., 2021a)) do not provide convergence rate analysis; while CAGrad requires that all task gradients are available in each epoch, which is very expensive (as will be demonstrated in Section 5.2).

## 5 EXPERIMENTS

In this section, we perform experiments on few-shot regression (Section 5.1), few-shot classification (Section 5.2), and reinforcement learning (Section 5.3). All experiments are run on a GeForce RTX 2080 Ti GPU and Intel(R) Xeon(R) CPU E5-2680. Our implementations are based on the popular open-source meta-learning library Learn2Learn (Arnold et al., 2020).

### 5.1 FEW-SHOT REGRESSION

**Setup**. We follow the setup in (Finn et al., 2017; Li et al., 2017). The target function for task $\tau$ is $y = a_\tau \sin(x + b_\tau)$, where $a_\tau$ and $b_\tau$ are sampled uniformly from $[0.1, 5.0]$ and $[0, \pi]$, respectively. We generate $160,000$ meta-training tasks and $1,000$ meta-testing tasks. A multilayer perceptron with 2 fully-connected (FC) layers (each of size 32) and ReLU activation is used as meta-learner and re-weighting network. The re-weighting network uses all mini-batch instances as input. To ensure that the re-weighting network output is positive, we take the square of its last layer's output as output.

The backbone meta-learning algorithm is MAML (Finn et al., 2017). We use Adam (Kingma & Ba, 2014), with an initial learning rate of $0.01$, to update the base learners for 5 steps in the inner-loop. For the outer-loop, we compare (i) minimizing the (single objective) of overall task loss as in MAML; versus performing MOO with (ii) mini-batch MGDA (Désidéri, 2012; Ye et al., 2021), (iii) mini-batch CAGrad, using the same hyper-parameters as in (Liu et al., 2021a), (iv) mini-batch PCGrad, using the

---

[4]The stop gradient operator satisfies $\perp (h(x)) = h(x)$, $\nabla_x \perp (h(\boldsymbol{x})) = 0$, where $h(\cdot)$ is any differentiable function.

same hyper-parameters as in (Yu et al., 2020), (v) the proposed SIMOL, and (vi) updating of (6) with a mini-batch version of the improvement function in (6). Hyperparameters for MAML, mini-batch MGDA, and mini-batch CAGrad follow (Finn et al., 2017), while that for the proposed SIMOL are in Appendix D.

The mini-batch size is 16. We do not compare with the batch versions of MGDA/CAGrad, as computing all task gradients takes very large memory and time. The initial learning rate for the outer loop is 0.001. The experiment is repeated three times with different random seeds. For performance evaluation, we use the mean-squared-error (MSE) over all meta-testing tasks. We also report the worst-10% MSE, which is the average MSE for the 10% worst-performing meta-testing tasks.

**Results**. Table 1 shows the MSE and its 95% confidence interval (computed as in (Finn et al., 2017; Li et al., 2017)). As can be seen, SIMOL consistently outperforms MAML, and the mini-batch versions of MGDA, CAGrad, PCGrad and improvement function in terms of both the overall and worst-10% MSEs. Indeed, the mini-batch versions of MGDA, CAGard, PCGrad and improvement function are even worse than the original MAML. Figure 2 shows the convergence of MSE with the number of training epochs. As can be seen, SIMOL converges slightly faster than the other baselines.

Table 1: MSE (with 95% confidence interval) for few-shot regression. The best results are in bold. The $^*$ denotes that the improvement over the second-best is statistically significant (at a significance level of 0.1 using the paired t-test).

|  | overall | | worst-10% | |
| --- | --- | --- | --- | --- |
|  | 5-shot | 2-shot | 5-shot | 2-shot |
| min average loss (MAML) | $0.43 \pm 0.11$ | $1.70 \pm 0.11$ | $2.13 \pm 0.21$ | $7.75 \pm 0.48$ |
| mini-batch MGDA | $0.60 \pm 0.02$ | $1.73 \pm 0.10$ | $2.62 \pm 0.10$ | $7.04 \pm 0.51$ |
| mini-batch CAGrad | $1.90 \pm 0.24$ | $1.82 \pm 0.44$ | $8.18 \pm 0.63$ | $8.23 \pm 2.33$ |
| mini-batch PCGrad | $1.89 \pm 0.12$ | $1.80 \pm 0.11$ | $6.64 \pm 0.47$ | $7.73 \pm 0.54$ |
| mini-batch improvement function | $0.69 \pm 0.03$ | $1.79 \pm 0.09$ | $2.63 \pm 0.21$ | $8.23 \pm 0.36$ |
| SIMOL | $\mathbf{0.34^* \pm 0.04}$ | $\mathbf{1.24^* \pm 0.08}$ | $\mathbf{1.69^* \pm 0.18}$ | $\mathbf{5.66^* \pm 0.33}$ |

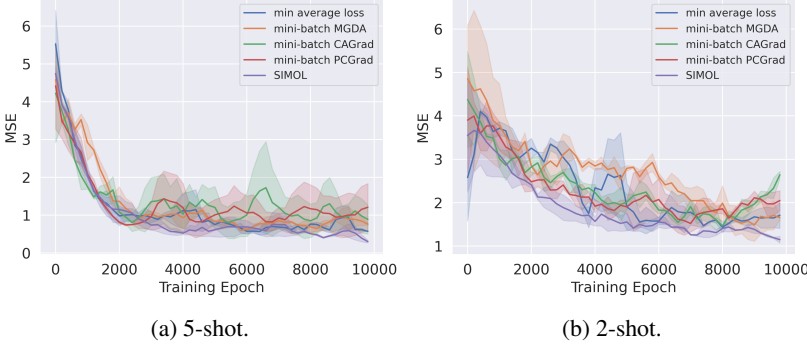

(a) 5-shot.          (b) 2-shot.

Figure 2: Convergence of MSE with the number of training epochs.

## 5.2 FEW-SHOT IMAGE CLASSIFICATION

**Setup**. In this section, we perform 5-way-1-shot and 5-way-5-shot classification on the *miniImageNet* (Ravi & Larochelle, 2016) and *tieredImageNet* data (Ren et al., 2018). Following (Finn et al., 2017), we split the *miniImageNet* dataset into a meta-training set with 64 classes, a meta-validation set with 16 classes, and a meta-testing set with 20 classes. The total number of meta-training tasks is $\binom{64}{5} \approx 7.6 \times 10^6$. Similarly, as in (Zhou et al., 2019), we split the *tieredImageNet* dataset into a meta-training set with 351 classes, a meta-validation set with 97 classes, and a meta-test set with 160 classes. The total number of meta-training tasks is $\binom{351}{5} \approx 4.3 \times 10^{10}$. For both datasets, we randomly select $1,000$ meta-testing tasks for evaluation.

We use two backbone meta-learning algorithms, MAML and prototypical network (PN) (Snell et al., 2017), with hyper-parameters following the original papers. Following (Finn et al., 2017; Li et al., 2017; Zintgraf et al., 2019), we use the CNN4[5] (LeCun et al., 2015) as the meta-learner, and a CNN4

---

[5]The CNN4 consists of four $3 \times 3$ convolution networks with batch normalization, $2 \times 2$ max-pooling and a ReLU activation layer.

with a 3-layer FC as the reweighting network. The optimizer is Adam. The learning rates for the meta networks are 0.003 for MAML-based methods and 0.001 for PN-based methods, respectively. The learning rate of the reweighting network is 0.08 for SIMOL and 0.0008 for SIMOL+PN. The mini-batch size is 32. More details on the hyperparameters are in Appendix D.

The proposed SIMOL is compared with (i) minimizing the overall task loss as in standard MAML, (ii) mini-batch MGDA, and (iii) mini-batch CAGrad. We also compare with MAML variants including (iv) Reptile (Nichol & Schulman, 2018), (v) FOMAML (Finn et al., 2017), (vi) Meta-MinibatchProx (Zhou et al., 2019), (vii) TSA-MAML (Zhou et al., 2021), (viii) IMAML (Rajeswaran et al., 2019)) (ix) MTL (Wang et al., 2021), a multi-task learning based maml approach, and the standard prototypical network (Snell et al., 2017). The evaluation metrics are similar to those in Section 5.1, but with accuracy instead of MSE. Experiments are repeated three times with different random seeds.

**Results**. Tables 2 and 3 show the meta-testing accuraccies and 95% confidence intervals on *miniImageNet* and *tieredImageNet*, respectively. For MAML and its variants, SIMOL consistently outperforms all the other baselines in terms of both the overall and worst-10% accuracies. The same is also observed on the meta-learning algorithm prototypical network. This demonstrates that SIMOL is useful for both gradient-based and metric-based meta-learning approaches.

Note that (mini-batch) MGDA and CAGrad do not perform good in terms of both overall and worst-10% accuracies, showing that they cannot be straightforwardly extended to the use of mini-batch. On the other hand, the batch versions of MGDA and CAGrad are computationally impractical. Table 4 compares the per-epoch running time in training stage of standard MAML, SIMOL, and batch MGDA and CAgrad. Experiment is performed on 5-way-5-shot classification with the MAML algorithm on *miniImageNet*. As can be seen, while SIMOL has comparable per-epoch running time as MAML, batch MGDA and CAgrad are much more computationally expensive (around $432,000$ times slower).

Table 2: 5-way classification accuracies on *miniImageNet* (with $95\%$ confidence interval). Results of Reptile, FOMAML, and Meta-MinibatchProx are from (Zhou et al., 2019), IMAML from (Deleu et al., 2022), and TSA-MAML from (Zhou et al., 2021). Results not reported in the original papers are denoted "-". The best results are in bold.

| | | overall | | worst-10% | |
|---|---|---|---|---|---|
| | | 1-shot | 5-shot | 1-shot | 5-shot |
| (MAML) | min average loss | $49.24 \pm 0.78$ | $62.13 \pm 0.72$ | $13.33 \pm 1.07$ | $41.71 \pm 1.02$ |
| | mini-batch MGDA | $46.08 \pm 0.78$ | $60.15 \pm 0.41$ | $10.60 \pm 1.33$ | $39.67 \pm 0.55$ |
| | mini-batch CAGrad | $44.67 \pm 0.75$ | $60.05 \pm 0.67$ | $11.33 \pm 1.12$ | $40.01 \pm 0.88$ |
| | SIMOL | **50.62\* $\pm$ 1.39** | **65.83\* $\pm$ 0.86** | **14.99\* $\pm$ 1.72** | **44.81\*$\pm$ 0.58** |
| (MAML variants) | Reptile | $47.07 \pm 0.26$ | $62.74 \pm 0.37$ | - | - |
| | FOMAML | $45.53 \pm 1.58$ | $61.02 \pm 1.12$ | - | - |
| | IMAML | $49.30 \pm 1.88$ | $59.77 \pm 0.41$ | - | - |
| | Meta-MinibatchProx | $48.51 \pm 0.92$ | $64.15 \pm 0.92$ | - | - |
| | TS-MAML | $48.44 \pm 0.91$ | $65.52 \pm 0.68$ | - | - |
| | MTL | $49.87 \pm 0.41$ | $65.81 \pm 0.33$ | $13.64 \pm 1.45$ | $43.42 \pm 0.47$ |
| (PN) | Standard | $48.25 \pm 0.95$ | $65.29 \pm 0.48$ | $13.10 \pm 1.32$ | $45.03 \pm 0.69$ |
| | SIMOL | **50.45\*$\pm$ 0.93** | **66.67\* $\pm$ 0.47** | **15.00\* $\pm$ 1.21** | **46.27\* $\pm$ 0.70** |

## 5.3 REINFORCEMENT LEARNING

In this experiment, experiments are performed on the *HalfCheetach-Dir* and *Walker-Dir* environments in Mujoco (Todorov et al., 2012). In both environments, each task corresponds to a random direction in the XY-plane, and the agent (Walker/ HalfCheetach) learns to run in that direction as far as possible. The reward is the average velocity minus control costs. We again use MAML as the meta-learning algorithm, and the base reinforcement learning algorithm is vanilla policy gradient (VPG) (Sutton et al., 1999). Following (Rothfuss et al., 2018), the policy network has two $64 \times 64$ FC layers with tanh activation, while the critic is a linear state-value function whose parameters are obtained by minimizing least-square. The re-weighting network has two $64 \times 64$ FC layers with ReLU activation. Following (Zintgraf et al., 2019), we use MAML as the baseline.

Figure 3 shows the convergence of the accumulated reward with the number of training iterations. As can be seen, SIMOL consistently outperforms MAML. Moreover, the convergence of MAML is less

Table 3: 5-way classification accuracies on *tieredImageNet* (with 95% confidence interval).

| | | overall | | worst-10% | |
| --- | --- | --- | --- | --- | --- |
| | | 1-shot | 5-shot | 1-shot | 5-shot |
| (MAML) | min average loss | $50.58 \pm 1.44$ | $69.33 \pm 0.74$ | $11.60 \pm 1.96$ | $46.95 \pm 1.14$ |
| | mini-batch MGDA | $22.92 \pm 1.04$ | $53.41 \pm 0.74$ | $7.12 \pm 2.11$ | $32.79 \pm 0.88$ |
| | mini-batch CAGrad | $49.04 \pm 0.93$ | $65.43 \pm 0.73$ | $11.40 \pm 1.97$ | $42.63 \pm 0.95$ |
| | SIMOL | **51.42* $\pm$ 1.50** | **70.13* $\pm$ 0.74** | **12.00* $\pm$ 1.95** | **47.51* $\pm$ 1.46** |
| (MAML variants) | Reptile | $49.12 \pm 0.43$ | $65.99 \pm 0.42$ | - | - |
| | FOMAML | $45.53 \pm 1.58$ | $61.02 \pm 1.12$ | - | - |
| | IMAML | $38.54 \pm 1.37$ | $60.24 \pm 0.76$ | - | - |
| | Meta-MinibatchProx | $50.14 \pm 1.37$ | $68.30 \pm 0.91$ | - | - |
| | TS-MAML | $48.82 \pm 0.88$ | $67.82 \pm 0.72$ | - | - |
| | MTL | $51.02 \pm 0.46$ | $66.47 \pm 0.39$ | $13.60 \pm 1.59$ | $49.45 \pm 0.58$ |
| (PN) | Standard | $47.63 \pm 0.93$ | $68.92 \pm 0.52$ | $14.40 \pm 1.25$ | $46.67 \pm 0.78$ |
| | SIMOL | **50.09* $\pm$ 0.96** | **71.51* $\pm$ 0.79** | **15.10* $\pm$ 1.20** | **49.85* $\pm$ 0.79** |

stable, as also reported by (Rothfuss et al., 2018). On the other hand, the convergence of SIMOL is smoother and more stable.

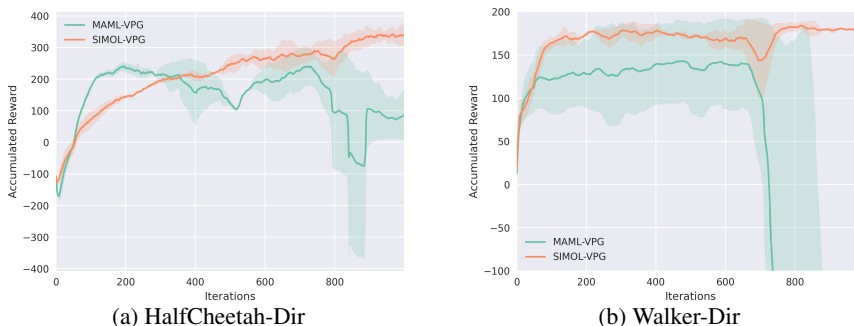

(a) HalfCheetah-Dir                   (b) Walker-Dir

Figure 3: Returns for SIMOL-VPG and MAML-VPG. Results are averaged over 3 trials.

## 5.4 ABLATION STUDY

In this experiment, we use the setup in Section 5.1, and vary the number of FC layers in SIMOL's meta-learner. The number of training epochs is always fixed to $10,000$. Table 5 shows the MSE's on 2-shot regression. as can be seen, the use of 2 FC layers has the best overall MSE and worst-10% MSE. the deeper networks may not be sufficiently trained with the fixed number of training epochs, leading to worse performance.

Table 4: Per-epoch running time for 5-way 5-shot classification on *miniImageNet*.

| MAML | SIMOL | batch MGDA | batch CAGrad |
| --- | --- | --- | --- |
| 2.0 sec | 2.3 sec | 6.9 days | 5.6 days |

Table 5: Performance for SIMOL with different numbers of layers in 2-shot regression.

| #layers | overall MSE | worst-10 % MSE |
| --- | --- | --- |
| 2 | $1.24 \pm 0.08$ | $5.66 \pm 0.33$ |
| 3 | $1.36 \pm 0.09$ | $6.01 \pm 0.38$ |
| 4 | $1.42 \pm 0.08$ | $6.22 \pm 0.34$ |

## 6 CONCLUSION

In this paper, we propose to avoid the compromising phenomenon in meta-learning by reformulating it as a multi-objective optimization (MOO) problem, in which each task is an objective. However, current gradient-based MOO solvers cannot scale to a large number of objectives. With the use of improvement function and mini-batch, we propose a scalable gradient-based solver with theoretical guarantees to Pareto-optimality. Empirical studies on few-shot regression, few-shot classification, and reinforcement learning demonstrate that the proposed method is efficient, and has good generalization in terms of both overall performance and performance on the poorly-performing tasks.

ACKNOWLEDGEMENT

This research was supported in part by the Research Grants Council of the Hong Kong Special Administrative Region (Grant 16200021).

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

## A  TOY EXAMPLE

The definitions of $f_1$ and $f_2$ are:

$$
\begin{aligned}
f_1(x) &= c_1(x)l_1(x) + c_2(x)g_1(x), \\
f_2(x) &= c_1(x)l_2(x) + c_2(x)g_2(x),
\end{aligned}
$$

where

$$
\begin{aligned}
l_1(x) &= \log\left(\max\left(\left|0.5\left(-x_1 - 7\right) - \tanh\left(-x_2\right)\right|, 0.000005\right)\right) + 6, \\
l_2(x) &= \log\left(\max\left(\left|0.5\left(-x_1 + 3\right) - \tanh\left(-x_2\right) + 2\right|, 0.000005\right)\right) + 6, \\
g_1(x) &= \left(\left(-x_1 + 7\right)^2 + 0.1 * \left(-x_2 - 8\right)^2\right)/10 - 20, \\
g_2(x) &= \left(\left(-x_1 - 7\right)^2 + 0.1 * \left(-x_2 - 8\right)^2\right)/10 - 20, \\
c_1(x) &= \max(\tanh(0.5x_2), 0), \\
c_2(x) &= \max(\tanh(-0.5x_2), 0).
\end{aligned}
$$

For mini-batch MGDA, and SIMOL the probability to sample task 1 is $2/3$ and $1/3$ for task 2. The learning rates for MGDA and mini-batch MGDA are both $0.01$. The learning rate for SIMOL's meta-learner is $0.01$, while that for its re-weighting network is $0.1$. Note that the meta-learner and the re-weighting network are represented by learnable vectors.

## B  PSEUDO-CODES

Algorithms 2 and 3 show the pseudo-codes for SIMOL-based MAML and PN, respectively. The key differences between SIMOL and MAML/PN are highlighted in blue.

---

**Algorithm 2:** SIMOL for MAML.

**Input:** $\mathcal{T}$, and total epoch $S$. $B$ is the batch size, $\beta$ and $\beta'$ are learning rates for $w$ and $\theta$. $\alpha$ is the learning rates for inner loop.

1 **for** *epoch* $s = 1, 2, 3, \ldots, S$ **do**
2     Sample tasks $1, 2, \ldots B$;
3     **for** *Every task* $\tau$ **do**
4        Receive $r_{\theta^s}(\tau)$;
5        Compute adapted parameters with gradient descent $w_\tau^*(w) = w^s - \alpha\nabla_w\mathcal{L}_\tau(w^s)$;
6     $w^{s+1} = w^s - \beta^s\frac{1}{B}\sum_{\tau=1}^{B} r_{\theta^s}(\tau)\nabla_w\mathcal{L}_\tau(w_\tau^*(w))$;
7     $\theta^{s+1} = \theta^s + \beta'^s\tilde{K}(\theta, B)$;

---

**Algorithm 3:** SIMOL for PN.

1 **for** *epoch* $s = 1, 2, 3, \ldots, S$ **do**
2     Sample tasks $1, 2, \ldots B$;
3     **for** *Every task* $\tau$ **do**
4        **for** *Every class* $c$ **do**
5           Select the support $S_\tau^c$ and query set $Q_\tau$;
6           Compute prototype $c_\tau = \frac{1}{N_C}\sum_{(\mathbf{x}_i, y_i)\in S_\tau^c} f_\theta(\mathbf{x}_i)$;
7        Calculate $w_\tau^*(w) = \frac{1}{|Q_\tau|N_C}\sum_{x\in Q_\tau}\frac{\exp(-\|f_w(x)-c_k\|^2)}{\sum_{k'}\exp(-\|f_w(x)-c_{k'}\|^2)}$;
8        Receive $r_{\theta^s}(\tau)$;
9     $w^{s+1} = w^s - \beta^s\frac{1}{B}\sum_{\tau=1}^{B} r_{\theta^s}(\tau)\nabla_w\mathcal{L}_\tau(w_\tau^*(w))$;
10     $\theta^{s+1} = \theta^s + \beta'^s\tilde{K}(\theta, B)$;

---

## C   PROOFS

### C.1   PROOF OF LEMMA 3.1

*Proof.* Note that

$$\max_{\tau=1\dots,m} \{\mathcal{L}_\tau(w_\tau^*(w)) - \mathcal{L}_\tau(w_\tau^*(w'))\} = \sum_\tau p(\tau)\mathcal{L}_\tau(w_\tau^*(w)) - \mathcal{L}_\tau(w_\tau^*(w')),$$

where

$$p(\tau) = \begin{cases} 1 & \tau = \arg\max_\tau \mathcal{L}_\tau(w_\tau^*(w)) - \mathcal{L}_\tau(w_\tau^*(w')) \\ 0 & \text{otherwise} \end{cases}.$$

First, we have

$$\max_{\pi(\tau)} \mathbb{E}_{\pi(\tau)}\left[\mathcal{L}_\tau(w_\tau^*(w)) - \mathcal{L}_\tau(w_\tau^*(w'))\right] - \sum_\tau p(\tau)\mathcal{L}_\tau(w_\tau^*(w)) - \mathcal{L}_\tau(w_\tau^*(w')) \geq 0.$$

This can be done by setting $\pi(\tau) = p(\tau)$. Next, we show that

$$\max_{\pi(\tau)} \mathbb{E}_{\pi(\tau)}\left[\mathcal{L}_\tau(w_\tau^*(w)) - \mathcal{L}_\tau(w_\tau^*(w'))\right] - \sum_\tau p(\tau)\mathcal{L}_\tau(w_\tau^*(w)) - \mathcal{L}_\tau(w_\tau^*(w')) \leq 0.$$

This is established since

$$\max_{\pi(\tau)} \mathbb{E}_{\tau\sim\pi(\tau)}\left[\mathcal{L}_\tau(w_\tau^*(w)) - \mathcal{L}_\tau(w_\tau^*(w'))\right] - \sum_\tau p(\tau)\mathcal{L}_\tau(w_\tau^*(w)) - \mathcal{L}_\tau(w_\tau^*(w'))$$

$$\leq \sum_\tau [\pi(\tau) - p(\tau)]\left[\mathcal{L}_\tau(w_\tau^*(w)) - \mathcal{L}_\tau(w_\tau^*(w'))\right]$$

$$= [\pi(\tau') - 1]\left[\mathcal{L}_{\tau'}(w_{\tau'}^*(w)) - \mathcal{L}_{\tau'}(w_{\tau'}^*(w'))\right] + \sum_{\tau\in\{1,\dots,m\}\setminus\tau'} [\pi(\tau)]\left[\mathcal{L}_\tau(w_\tau^*(w)) - \mathcal{L}_\tau(w_\tau^*(w'))\right]$$

$$\overset{(a)}{\leq} [\pi(\tau') - 1]\left[\mathcal{L}_{\tau'}(w_{\tau'}^*(w)) - \mathcal{L}_{\tau'}(w_{\tau'}^*(w'))\right] + \sum_{\tau\in\{1,\dots,m\}\setminus\tau'} [\pi(\tau)]\left[\mathcal{L}_{\tau'}(w_{\tau'}^*(w)) - \mathcal{L}_{\tau'}(w_{\tau'}^*(w'))\right]$$

$$\leq \pi(\tau')\left[\mathcal{L}_{\tau'}(w_{\tau'}^*(w)) - \mathcal{L}_{\tau'}(w_{\tau'}^*(w'))\right] + \sum_{\tau\in\{1,\dots,m\}\setminus\tau'} [\pi(\tau)]\left[\mathcal{L}_{\tau'}(w_{\tau'}^*(w)) - \mathcal{L}_{\tau'}(w_{\tau'}^*(w'))\right]$$

$$- \left[\mathcal{L}_{\tau'}(w_{\tau'}^*(w)) - \mathcal{L}_{\tau'}(w_{\tau'}^*(w'))\right]$$

$$= \left[\mathcal{L}_{\tau'}(w_{\tau'}^*(w)) - \mathcal{L}_{\tau'}(w_{\tau'}^*(w'))\right] - \left[\mathcal{L}_{\tau'}(w_{\tau'}^*(w)) - \mathcal{L}_{\tau'}(w_{\tau'}^*(w'))\right]$$

$$= 0,$$

where $\tau' = \arg\max_{\tau=1\dots,m} \{\mathcal{L}_\tau(w_\tau^*(w)) - \mathcal{L}_\tau(w_\tau^*(w'))\}$. (a) is due to the fact that $\left[\mathcal{L}_{\tau'}(w_{\tau'}^*(w)) - \mathcal{L}_{\tau'}(w_{\tau'}^*(w'))\right] \geq \left[\mathcal{L}_\tau(w_\tau^*(w)) - \mathcal{L}_\tau(w_\tau^*(w'))\right], \forall\tau$ based on the property of $\tau'$. Therefore, we have

$$\left[\mathcal{L}_{\tau'}(w_{\tau'}^*(w)) - \mathcal{L}_{\tau'}(w_{\tau'}^*(w'))\right] \geq \left[\mathcal{L}_\tau(w_\tau^*(w)) - \mathcal{L}_\tau(w_\tau^*(w'))\right]$$

and

$$\left[\mathcal{L}_{\tau'}(w_{\tau'}^*(w)) - \mathcal{L}_{\tau'}(w_{\tau'}^*(w'))\right] \leq \left[\mathcal{L}_\tau(w_\tau^*(w)) - \mathcal{L}_\tau(w_\tau^*(w'))\right].$$

Thus,

$$\max_{\tau=1\dots,m} \{\mathcal{L}_\tau(w_\tau^*(w)) - \mathcal{L}_\tau(w_\tau^*(w'))\} = \max_{\pi(\tau)} \mathbb{E}_{\tau\sim\pi(\tau)}\left[\mathcal{L}_\tau(w_\tau^*(w)) - \mathcal{L}_\tau(w_\tau^*(w'))\right].$$

□

### C.2   PROOF OF PROPOSITION 3.2

*Proof.* Using the first-order Taylor expansion,

$$\arg\min_d \mathbb{E}_{U(\tau)}[r_\theta(\tau)[\mathcal{L}_\tau(w_\tau^*(w^s + d)) - (\mathcal{L}_\tau(w_\tau^*(w^s)))] + \frac{\lambda'}{2}\|d\|^2$$

$$= \arg\min_d \mathbb{E}_{U(\tau)}\left[r_\theta(\tau)\left[\nabla_w \mathcal{L}_\tau(w_\tau^*(w^s))^\top d\right]\right] + \frac{\lambda'}{2}\|d\|^2. \tag{17}$$

Taking the derivatives w.r.t. $d$,

$$\nabla_d \mathbb{E}_{U(\tau)}\left[r_\theta(\tau)\left[\nabla_w \mathcal{L}_\tau(w_\tau^*(w^s))^\top d\right]\right] + \frac{\lambda'}{2}\|d\|^2 = \mathbb{E}_{U(\tau)}[r_\theta(\tau)[\nabla_w \mathcal{L}_\tau(w_\tau^*(w^s))]] + \lambda' d.$$

Setting the above to zero, we have $\mathbb{E}_{U(\tau)}[r_\theta(\tau)\nabla_w \mathcal{L}_{D_\tau}(w_\tau^*(w^s))] + d = 0$, and

$$d^* = \frac{-1}{\lambda'}\mathbb{E}_{U(\tau)}[r_\theta(\tau)[\nabla_w \mathcal{L}_\tau(w_\tau^*(w^s))]].$$

Putting $d^*$ back to the objective in (17), we have:

$$\mathbb{E}_{\tau\sim U}r_\theta(\tau)[\mathcal{L}_\tau(w_\tau^*(w^s + d)) - \mathcal{L}_\tau(w_\tau^*(w^s))] + \frac{\lambda'}{2}\|d\|^2$$

$$= \left[\mathbb{E}_{\tau\sim U}r_\theta(\tau)\nabla_w \mathcal{L}_\tau(w_\tau^*(w))|_{w=w^s}^\top d\right] + \frac{\lambda'}{2}\left\|-\frac{1}{\lambda'}\mathbb{E}_{\tau\sim\pi}[\nabla_w \mathcal{L}_\tau(w_\tau^*(w)|_{w=w^s}]\right\|^2$$

$$= \left\langle\mathbb{E}_{\tau\sim U}r_\theta(\tau)\nabla_w \mathcal{L}_\tau(w_\tau^*(w))|_{w=w^s}^\top, d\right\rangle + \frac{1}{2\lambda'}\left\|\mathbb{E}_{\tau\sim\pi}[\nabla_w \mathcal{L}_\tau(w_\tau^*(w)|_{w=w^s}]\right\|^2$$

$$= -\frac{1}{\lambda'}\left\langle\mathbb{E}_{\tau\sim U}r_\theta(\tau)\nabla_w \mathcal{L}_\tau(w_\tau^*(w))|_{w=w^s}^\top, \mathbb{E}_{\tau\sim U}r_\theta(\tau)\nabla_w \mathcal{L}_\tau(w_\tau^*(w))|_{w=w^s}^\top\right\rangle$$

$$+\frac{1}{2\lambda'}\|\mathbb{E}_{\tau\sim\pi}[\nabla_w \mathcal{L}_\tau(w_\tau^*(w)|_{w=w^s}]\|^2$$

$$= \frac{-1}{\lambda'}\|\mathbb{E}_{\tau\sim\pi}[\nabla_w \mathcal{L}_\tau(w_\tau^*(w)|_{w=w^s}]\|^2.$$

$\square$

Next, we have the following two Lemmas.

**Lemma C.1.** *When $\mathcal{L}_\tau(w_\tau^*(w^s + d)) \approx \mathcal{L}_\tau(w_\tau^*(w^s) + \nabla_w \mathcal{L}_\tau(w_\tau^*(w^s))^\top d$, Eq. (9) is convex w.r.t. $d$ and concave. w.r.t. $r_\theta(\tau)$.*

*Proof.* Putting $\mathcal{L}_\tau(w_\tau^*(w^s + d)) \approx \mathcal{L}_\tau(w_\tau^*(w^s) + \nabla_w \mathcal{L}_\tau(w_\tau^*(w^s))^\top d$ into Eq. (9), we have:

$$\min_d \max_{r(\tau)} \mathbb{E}_{U(\tau)}[r(\tau)[\mathcal{L}_\tau(w_\tau^*(w^s + d)) - (\mathcal{L}_\tau(w_\tau^*(w^s)))] + \frac{\lambda'}{2}\|d\|^2 - \frac{\lambda''}{2}(\mathbb{E}_{U(\tau)}[r(\tau)] - 1)^2]$$

$$= \min_d \max_{r(\tau)} \mathbb{E}_{U(\tau)}[r(\tau)[\mathcal{L}_\tau(w_\tau^*(w^s)) + \nabla_w \mathcal{L}_\tau(w_\tau^*(w^s))^\top d - (\mathcal{L}_\tau(w_\tau^*(w^s)))]$$

$$+\frac{\lambda'}{2}\|d\|^2 - \frac{\lambda''}{2}(\mathbb{E}_{U(\tau)}[r(\tau)] - 1)^2].$$

For $\nabla_w \mathcal{L}_\tau(w_\tau^*(w^s))^\top d + \frac{\lambda'}{2}\|d\|^2$, since $\nabla_w \mathcal{L}_\tau(w_\tau^*(w^s))^\top d$ is both convex and concave w.r.t. $d$, and $\frac{\lambda'}{2}\|d\|^2$ is convex w.r.t. $d$. Then, $\nabla_w \mathcal{L}_\tau(w_\tau^*(w^s))^\top d + \frac{\lambda'}{2}\|d\|^2$ is convex w.r.t $d$. Thus, the sum of two convex functions $\nabla_w \mathcal{L}_\tau(w_\tau^*(w^s))^\top d + \frac{\lambda'}{2}\|d\|^2$ is convex w.r.t. $d$. Thus, Eq. (9) is convex w.r.t. $d$. Similarly, since $-\frac{\lambda''}{2}(\mathbb{E}_{U(\tau)}[r_\theta(\tau)] - 1)^2$ is concave w.r.t. $r_\theta(\tau)$. Therefore, Eq. (9) is also concave w.r.t. $r_\theta(\tau)$. $\square$

**Lemma C.2.** *When $\mathcal{L}_\tau(w_\tau^*(w^s + d)) \approx \mathcal{L}_\tau(w_\tau^*(w^s) + \nabla_w \mathcal{L}_\tau(w_\tau^*(w^s))^\top d$,*

$$\min_d \max_{r_\theta(\tau)} \mathbb{E}_{U(\tau)}[r_\theta(\tau)[\mathcal{L}_\tau(w_\tau^*(w^s + d)) - \mathcal{L}_\tau(w_\tau^*(w^s))] + \frac{\lambda'}{2}\|d\|^2 - \frac{\lambda''}{2}(\mathbb{E}_p[r_\theta(\tau)] - 1)^2]$$

$$= \max_{r_\theta(\tau)} \min_d \mathbb{E}_{U(\tau)}[r_\theta(\tau)[\mathcal{L}_\tau(w_\tau^*(w^s + d)) - \mathcal{L}_\tau(w_\tau^*(w^s))] + \frac{\lambda'}{2}\|d\|^2 - \frac{\lambda''}{2}(\mathbb{E}_p[r_\theta(\tau)] - 1)^2].$$

*Proof.* The above equation is established by the minimax theorem (Simons, 1995) when Eq. (9) is convex w.r.t. $d$ and concave w.r.t. $r_\theta(\tau)$. This holds by using Lemma C.1. $\square$

**Lemma C.3.** *If every $B \in \mathcal{B}$ is a set consists of a unique selection of $k$ ($m > k > 1$) tasks out of $m$ (the total number of tasks) tasks without replacement. For any continuous function $f(\cdot)$, we have:*

$$\left[\frac{1}{m}\sum_{\tau=1}^{m} f(\tau)\right]^2 = C_1 \sum_{B \in \mathcal{B}}\left[\sum_{\tau \in B} f(\tau)\right]^2 - C_2 \sum_{B \in \mathcal{B}}\sum_{\tau \in B}[f(\tau)]^2,$$

*where $C_1 = \frac{k^2}{\binom{m-2}{k-2}m^2}$, $C_2 = \left[\binom{m-1}{k-1} - \frac{1}{2}\binom{m-2}{k-2}\right]/\binom{m-1}{k-1}m^2\binom{m-2}{k-2}$.*

*Proof.* Note that

$$\left[\frac{1}{m}\sum_{\tau=1}^{m} f(\tau)\right]^2$$

$$\overset{(a)}{=} \sum_{\tau'}\sum_{\tau} \mathbb{1}(\tau, \tau') f(\tau) f(\tau')$$

$$\overset{(b)}{=} \frac{1}{\binom{n-2}{B-2}} \sum_{B \in \mathcal{B}}\sum_{\tau' \in B}\sum_{\tau \in B} \mathbb{1}(\tau, \tau') f(\tau) f(\tau') - \left[\binom{n-1}{B-1} - \frac{1}{2}\binom{n-2}{B-2}\right]\sum_{\tau \in \mathcal{T}} f(\tau)^2$$

$$= \frac{1}{\binom{n-2}{B-2}}\left[\sum_{B \in \mathcal{B}}\sum_{\tau' \in B}\sum_{\tau \in B} \mathbb{1}(\tau, \tau') f(\tau) f(\tau') - \frac{\left[\binom{n-1}{B-1} - \frac{1}{2}\binom{n-2}{B-2}\right]}{\binom{n-1}{B-1}}\sum_{B \in \mathcal{B}}\sum_{\tau \in B} f(\tau)^2\right]$$

$$= \frac{1}{\binom{n-2}{B-2}m^2}\left[\sum_{B \in \mathcal{B}}[\sum_{\tau \in B} f(\tau)]^2 - \frac{\left[\binom{n-1}{B-1} - \frac{1}{2}\binom{n-2}{B-2}\right]}{\binom{n-1}{B-1}m^2}\sum_{B \in \mathcal{B}}\sum_{\tau \in B} f(\tau)^2\right]$$

$$= C_1[\sum_{B \in \mathcal{B}}\left[\sum_{\tau \in B} f(\tau)\right]^2 - C_2 \sum_{B \in \mathcal{B}}\sum_{\tau \in B}[f(\tau)]^2.$$

(a) is due to the multinomial theorem (Bolton, 1968) and $\mathbb{1}(\tau, \tau') = \begin{cases} 2 & \tau \neq \tau' \\ 1 & \text{otherwise} \end{cases}$. (b) is due to the fact that every task $\tau$ has occurred exactly $\binom{n-1}{B-1}$ times, and every tuple $(\tau, \tau')$ has occurred exactly $\binom{n-2}{B-2}$ times. $\qquad\square$

**Lemma C.4.** *If every $B \in \mathcal{B}$ is a set consists of a unique selection of $k$ ($m > k > 1$) tasks out of $m$ (the total number of tasks) tasks without replacement. Let $g(\tau) = [g_1(\tau), g_2(\tau), \ldots, g_m(\tau)]$, where $g_i(\cdot)$'s are continuous functions. We have*

$$\left\|\frac{1}{m}\sum_{\tau=1}^{m} g(\tau)\right\|^2 = C_1 \sum_{B \in \mathcal{B}}\left\|\sum_{\tau \in B} g(\tau)\right\|^2 - C_2 \sum_{B \in \mathcal{B}}\sum_{\tau \in B}\|g(\tau)\|^2,$$

*where $C_1 = \frac{k^2}{\binom{m-2}{k-2}m^2}$, and $C_2 = \left[\binom{m-1}{k-1} - \frac{1}{2}\binom{m-2}{k-2}\right]/\binom{m-1}{k-1}m^2\binom{m-2}{k-2}$.*

*Proof.* Observe that $\|g(\tau)\|^2 = \sum_i g_i(\tau)^2$. The remaining follows from Lemma C.3.

$\qquad\square$

*Proof.* (of Proposition 3.3) Using Lemmas C.3 and C.4, we have:

$$\tilde{K}_{\mathcal{T}}(\theta) - \frac{1}{|\mathcal{B}|}\sum_B \tilde{K}_B(\theta)$$

$$= \sum_{B\in\mathcal{B}}\left[-\frac{3C_1}{2\lambda'} + \frac{3}{2\lambda'|\mathcal{B}|}\right]||\tilde{d}^*||^2 - \sum_{B\in\mathcal{B}}\left[\lambda''C_1 - \frac{\lambda''}{|\mathcal{B}|k^2}\right]\left(\sum_{\tau\in B}(r_\theta(\tau)-1)\right)^2$$

$$-C_2\left[\sum_{B\in\mathcal{B}}\sum_{\tau\in B}\left(-\frac{1}{\hat{\lambda}'}||r_\theta(\tau)\nabla_w\mathcal{L}_\tau(w^*_\tau(w))|_{w=w^s}||^2 - \hat{\lambda}''[r_\theta(\tau)-1]^2\right)\right]$$

$$= \left[\sum_{B\in\mathcal{B}}\sum_{\tau\in B}\left(-C_2\frac{1}{\lambda'}||r_\theta(\tau)\nabla_w\mathcal{L}_\tau(w^*_\tau(w))|_{w=w^s}||^2 - C_2\lambda''[r_\theta(\tau)-1]^2\right)\right],$$

where $\tilde{d}^* := \frac{1}{|B|}\sum_{B\in\mathcal{B}} r_\theta(\tau)\nabla_w\mathcal{L}_\tau(w^*_\tau(w))|_{w=w^s}$. Recall $\hat{\lambda}'$ and $\hat{\lambda}''$ are defined in Sec. 3. Then,

$$(\tilde{K}_{\mathcal{T}}(\theta) - \frac{1}{|\mathcal{B})}\sum_B \tilde{K}_B(\theta))^2 \tag{18}$$

$$\leq \left(\left[\left(-\frac{C_2k|\mathcal{B}|}{\lambda'}\frac{1}{k|\mathcal{B}|}\sum_{B\in\mathcal{B}}\sum_{\tau\in B}||r_\theta(\tau)\nabla_w\mathcal{L}_\tau(w^*_\tau(w))|_{w=w^s}||^2\right.\right.\right. \tag{19}$$

$$\left.\left.\left.+C_2k|\mathcal{B}|\lambda''\frac{1}{k|\mathcal{B}|}\sum_{B\in\mathcal{B}}\sum_{\tau\in B}[r_\theta(\tau)-1]^2\right)\right]\right)^2 \tag{20}$$

$$\leq \frac{C_2k|\mathcal{B}|G_1}{\lambda'} + C_2k|\mathcal{B}|\lambda''G_2. \tag{21}$$

By noting that $\tilde{K}_{\mathcal{T}}(\theta)$ is exactly $K(\theta)$, we obtain the first claim.

Regarding the second claim, note that when $m$ is large,

$$C_2k|\mathcal{B}| = \frac{k\left[\binom{m-1}{k-1} - \frac{1}{2}\binom{m-2}{k-2}\right]\binom{m}{k}}{\binom{m-1}{k-1}m^2\binom{m-2}{k-2}}$$

$$\approx \frac{k\binom{m-1}{k-1}\binom{m}{k}}{\binom{m-1}{k-1}m^2\binom{m-2}{k-2}} = \frac{k\binom{m}{k}}{m^2\binom{m-2}{k-2}}$$

$$= \frac{k\binom{m-1}{k-1}\frac{m}{k}}{m^2\binom{m-2}{k-2}} = \frac{k\binom{m-2}{k-2}\frac{m}{k}\frac{m-1}{k-1}}{m^2\binom{m-2}{k-2}}$$

$$= \frac{m-1}{(k-1)m} \approx \frac{1}{k-1}.$$

The first approximation is due to the fact that $\binom{m-1}{k-1} \gg \frac{1}{2}\binom{m-2}{k-2}$. The third and forth equations are due to the property of combinatorics ($\binom{m}{k} = \frac{m}{k}\binom{m-1}{k-1}$). Putting $\frac{1}{k-1}$ back into Eq. (18), we obtain the results. $\square$

## C.3 PROOF OF LEMMA 4.1

*Proof.* We have:

$$\nabla_\theta R(\theta, w) = \nabla_\theta\Delta(\theta) - \nabla_\theta\frac{\lambda''}{2}(\mathbb{E}_{\tau\sim U}r_\theta(\tau)-1)^2$$

$$= -\frac{1}{\lambda'}\mathbb{E}_{\tau\sim U}\nabla_\theta r_\theta(\tau)\perp[(\mathcal{L}_\tau(w^*_\tau(w)+d^*) - \mathcal{L}_\tau(w^*_\tau(w))]$$

$$\perp[\mathbb{E}_{\tau\sim U}r_\theta(\tau)[\mathcal{L}_\tau(w^*_\tau(w)+d^*) - \mathcal{L}_\tau(w^*_\tau(w))]]$$

$$= \nabla_\theta||\mathbb{E}_{\tau\sim U}\nabla_\theta r_\theta(\tau)\nabla_w\mathcal{L}_\tau(w^*_\tau(w))|_{w=w^s}||^2 - \nabla_\theta\frac{\lambda''}{2}(\mathbb{E}_{\tau\sim U}r_\theta(\tau)-1)^2$$

$$= \nabla_\theta K(\theta).$$

Also notice that:

$$\nabla_w R(\theta, w)$$
$$= \mathbb{E}_{\tau \sim U}[\perp (r_\theta(\tau)) [\nabla_w \mathcal{L}_\tau(w_\tau^*(w))] + \nabla_w \Delta(\theta) - \nabla_w \frac{\lambda''}{2}(\mathbb{E}_{\tau \sim U} r_\theta(\tau) - 1)^2]$$
$$= \mathbb{E}_{\tau \sim U} r_\theta(\tau)[\nabla_w L_\tau(w_\tau^*(w))].$$

Therefore, $\nabla_w R(\theta, w)|_{w=w_s} = -d^*$. $\qquad\square$

## C.4 Proof of Theorem 4.2

**Definition C.5.** *(Weak Global Pareto Optimality) (Mäkelä et al., 2016) A solution $x^*$ of problem (5) is weakly global Pareto optimal if there does not exist another $x$ such that $f_\tau(x^*) \geq f_\tau(x)$ for $\tau \in \{1, \ldots, m\}$.*

**Theorem C.6.** *(Theorem 5 in (Miettinen & Mäkelä, 1995)) For a multi-objective optimization problem $\min_w [f_1(w), f_2(w), \ldots, f_m(w)]$ and its corresponding improvement function $H(w, w^*)$. A necessary condition for $w^* \in \mathbb{R}^n$ to be weakly global Pareto optimal is that $w^* = \arg\min_w H(w, w^*)$. Moreover, if $f_i(w)$ is convex $\forall i$, then it is a sufficient condition.*

We observe the following.

1) By replacing $f_\tau = \mathcal{L}_\tau \circ w_\tau^*(w), \forall \tau$, the above theorem can be directly applied to the MAML setting.

2) $f$ in our setting can be a neural network. Thus, the convexity condition for $f_\tau(w)$ can be hard to satisfy. Here, we give the a more relaxed version of the above theorem:

**Theorem C.7.** $w^* := \arg\min_w H(w, w^*)$ *is Pareto stationary.*

*Proof.* Using Theorem 2 in (Miettinen & Mäkelä, 1995), we have

$\partial_w H(w, w^*) \subset \operatorname{conv} \bigcup_i \partial_w (f_i(w) - f_i(w^*))$, where $\partial$ is reloaded as sub-gradient, and $\operatorname{conv}$ is the convex set.

Note that $0 \in \partial_w H(w, w^*)$ due to the fact that the element in the convex set of the union of sub-gradients is still a sub-gradient, and for $w \in \{w|0 \in \partial_w H(w, w^*)\}$, the sub-gradient $\partial_w H(w, w^*)$ is zero.

Then, we always have $\sum_i w_i \partial_w (f_i(w) - f_i(w^*)) = 0$, $\sum_i w_i = 1$, based by definition of a convex set, where $w_i \in [0, 1], \forall i$ is a real number. By simplifying the above term, we have $\sum_i w_i \partial (f_i(w) - f_i(w^*)) = \sum_i w_i \partial_w f_i(w) = 0$. which is exactly the definition of Pareto stationary point. $\qquad\square$

We now show convergence of Algorithm 1.

**Lemma C.8.** *In each epoch $s$ of Algorithm 1, set $\hat{\lambda}'' = \begin{cases} 0 & \mathbb{E}_p[r_\theta(\tau)] = 1 \\ \infty & otherwise \end{cases}$, we have:*

$$H(w^s + d, w^s) \leq H(w^s, w^s).$$

*When equality holds, $w^s$ is a stationary point.*

*Proof.* By Lemma C.2, the solutions of

$$\max_\theta \min_d \mathbb{E}_{U(\tau)}[r_\theta(\tau) [\mathcal{L}_\tau(w_\tau^*(w^s + d)) - \mathcal{L}_\tau(w_\tau^*(w^s))] + \tfrac{\hat{\lambda}'}{2}||d||^2 - \tfrac{\hat{\lambda}''}{2}(\mathbb{E}_p[r_\theta(\tau)] - 1)^2]$$

are the same as those from

$$\min_d \max_\theta \mathbb{E}_{U(\tau)}[r_\theta(\tau) [\mathcal{L}_\tau(w_\tau^*(w^s + d)) - \mathcal{L}_\tau(w_\tau^*(w^s))] + \tfrac{\hat{\lambda}'}{2}||d||^2 - \tfrac{\hat{\lambda}''}{2}(\mathbb{E}_p[r_\theta(\tau)] - 1)^2].$$

Solving $\theta$, the above minimax problem degenerates to

$$\min_d \max_{\tau=1\ldots,m} [\mathcal{L}_\tau(w_\tau^*(w^s + d)) - \mathcal{L}_\tau(w_\tau^*(w^s))] + \frac{\hat{\lambda}'}{2}||d||^2.$$

Note that the above optimization problem is exactly $\min_d H(w^s + d, w^s) + \frac{\hat{\lambda}'}{2}||d||^2$, where $d$ is the steepest descent direction. Then, due to the property of steepest descent direction, we always have $H(w^s + d, w^s) \leq H(w^s, w^s)$, and equality holds iff $||d||^2 = 0$, where $w^s$ a stationary point according to the definition. $\square$

**Lemma C.9.** *Algorithm 1 converges to a Pareto stationary point. Moreover, if $\mathcal{L}_\tau(w_\tau^*(w))$ is convex, Algorithm 1 converges to a global weak Pareto optimal point.*

*Proof.* From Lemma C.8, the sequence $\{H(w^s + d, w^s, w^{s+1}_{1:m}, w^s_{1:m})\}_s$ is decreasing. Also obviously, $H(w^s + d, w^s, w^{s+1}_{1:m}, w^s_{1:m})$ has a lower bound. Thus, using the monotone convergence theorem, Algorithm 1 converges. Using Theorem C.7, it is Pareto stationary. Together with Theorem C.6, it is global Pareto optimal if $\mathcal{L}_\tau(w_\tau^*(w))$ is convex. $\square$

**Lemma C.10.** *(i) If $\mathcal{L}_\tau(w_\tau^*(w + d))$ is L-smooth and $\mu'$-convex w.r.t. $w$ and $d$, and $r_\theta(\tau)$ is $\mu$-concave, $r_\theta(\tau)$ and $\mathcal{L}_\tau(w_\tau^*(w^{s+1}))$ are bounded (i.e., $r_\theta(\tau) \leq C_{\max}$ and $\mathcal{L}_\tau(w_\tau^*(w^{s+1})) \leq C'_{\max}$), then $R(\theta, w)$ is $L + C_{\max}\mu'$ convex w.r.t. $w$ and 2L-smooth w.r.t. $w, \theta$ and $C'_{\max}\mu$ concave w.r.t. $\theta$.*

*(ii) If $\mathcal{L}_\tau(w_\tau^*(w + d))$ is L-smooth w.r.t. $w$ and $d$, and $r_\theta(\tau)$ is $\mu$-concave, then $\tilde{R}(\theta, w, B)$ is L-smooth w.r.t. $w, \theta$, and $C'_{\max}\mu$-concave w.r.t. $\theta$.*

*Proof.* For claim (i), we first prove that for any differentiable functions $f, g, \perp [f(x)] \perp [g(x)]$ is 0-smooth and convex w.r.t. $x$. Note that

$$||\nabla_x \perp [f(x)]\perp [g(x)] - \nabla_{x'} \perp [f(x')]\perp [g(x')] \; ||^2 = 0$$

due to the property of the stop gradient operator. Thus,

$$\perp [f(x)]\perp [g(x)]- \perp [f(x')]\perp [g(x')] = -\nabla_{x'}(\perp [f(x')]\perp [g(x')])(\perp [f(x)] \perp [g(x)]) = 0.$$

For convexity, by using the property of 0-smoothness, we have

$$\perp [f(x')]\perp [g(x')]- \perp [f(x)]\perp [g(x)] \geq \nabla_{x'}(\perp [f(x')]\perp [g(x')])(\perp [f(x)]\perp [g(x)]).$$

Thus, $\perp [f(x)] \perp [g(x)]$ is convex due to the definition of convexity. Therefore, $\perp [f(x)] \perp [g(x)]$ is also 0-smooth and convex w.r.t. $x$.

A direct application of the above, we obtain $\perp [f(x)]$ is 0-smooth and convex w.r.t. $x$ by setting $g(x) \equiv 1$.

Using the above, we have $\perp \; [(L_\tau(w_\tau^*(w) + d) - \mathcal{L}_\tau(w_\tau^*(w))], \perp \; [\mathbb{E}_{\tau\sim U}r_\theta(\tau)[\mathcal{L}_\tau(w_\tau^*(w) + d^*) - \mathcal{L}_\tau(w_\tau^*(w))]], \perp \; [(\mathcal{L}_{D_T}(w_T^*(w) + d^*) - \mathcal{L}_{D_T}(w_T^*(w))]] \perp \mathbb{E}_{T\sim U}r_\theta(T)[\mathcal{L}_{D_T}(w_T^*(w) + d^*) - \mathcal{L}_{D_T}(w_T^*(w))]$ are also 0-smooth and convex. Therefore $r_\theta(T) \perp [(\mathcal{L}_{D_T}(w_T^*(w) + d^*) - \mathcal{L}_{D_T}(w_T^*(w))]] \perp [\mathbb{E}_{T\sim U}r_\theta(T)[\mathcal{L}_{D_T}(w_T^*(w) + d^*) - \mathcal{L}_{D_T}(w_T^*(w))]]$ is convex and L-smooth (as $L \geq 0$).

Also, we have $\perp (r_\theta(\tau))[\mathcal{L}_\tau(w_\tau^*(w))]$ is $C_{\max}\mu'$-convex and L-smooth due to our assumption.

Then, $\mathbb{E}_{\tau\sim U}[\perp (r_\theta(\tau))[\mathcal{L}_\tau(w_\tau^*(w))]$ is $C_{\max}\mu'$-convex and L-smooth since the addition of convex functions implies $\mathbb{E}_{\tau\sim U}[\perp (r_\theta(\tau))[\mathcal{L}_\tau(w_\tau^*(w))]$ is $C_{\max}\mu'$-convex, and the adding of smooth functions implies $\mathbb{E}_{\tau\sim U}[\perp (r_\theta(\tau))[\mathcal{L}_\tau(w_\tau^*(w))]$ is L-smooth, and taking the average does not affect the results. Thus, $R(\theta, w)$ is $C_{\max}\mu'$-convex and L-smooth.

For $r_\theta$, since $-\frac{\lambda'}{2}\nabla_\theta(\mathbb{E}_p[r_{\theta^s}(\tau)] - 1)^2 + \Delta(\theta)$ is concave, $\Delta(\theta)$ is $2\mu C'_{\max}$-concave. Then $\Delta(\theta) - \frac{\lambda''}{2}(\frac{1}{B}\sum_\tau[r_\theta(\tau)] - 1)^2$ is also $C'_{\max}\mu$-concave. Thus $R(\theta, w)$ is $C'_{\max}\mu$-concave.

The proof of claim (ii) is similar. $\square$

*Proof.* (of Theorem 4.2) Recall Lemma C.10 on the properties of convex and smooth for $\tilde{R}$. Combine it with the assumptions in Theorem 4.2, and use Theorem 4.9 in (Lin et al., 2020), we obtain the bound of $O(\frac{1}{\epsilon^8})$ when $B = 1$.

When $B > 1$, note that using Lemma A.2 in (Lin et al., 2020), we have:

$$\mathbb{E}\left[\left\|\frac{1}{B}\sum_{i=1}^B \tilde{R}(\theta, w, B)\right\|^2\right] \leq \|\nabla_w R(\theta, w)\|^2 + \frac{\sigma^2}{B}.$$

Therefore, let $\sigma^{2'} = \frac{\sigma^2}{B}$ and use Theorem 4.9 in (Lin et al., 2020) shows the bound of $O(\frac{1}{\epsilon^8})$.

The remaining part that the fixed point $w$ of $\max_\theta \tilde{R}(\theta, w, B)$ is global Pareto optimal (resp. Pareto stationary) can be obtained by using Lemma C.9, which says that the fixed point of Algorithm 1 is global Pareto optimal (resp. Pareto stationary).

$\square$

Finally, we show that using SIMOL on MAML can guarantee convergence.

*Proof.* (of Corollary 4.2.1) First, we show that if $\nabla_w \mathcal{L}_\tau(w)$ is Hessian-Lipschitz continuous, bounded and Lipschitz-continuous, and $\|w\|$ is bounded, then $w_\tau^*(w)$ is also Hessian-Lipschitz continuous, bounded and Lipschitz-continuous. Note that

$$
\begin{aligned}
\|\nabla_w^2 w_\tau^*(w) - \nabla_{w'}^2 w_\tau^*(w')\| &= \alpha \|\nabla_w^2 (\nabla_w \mathcal{L}_\tau(w)) - \nabla_{w'}^2 (\nabla_{w'} \mathcal{L}_\tau(w'))\|, \\
\|\nabla_w w_\tau^*(w) - \nabla_{w'} w_\tau^*(w')\| &= \alpha \|\nabla_w (\nabla_w \mathcal{L}_\tau(w)) - \nabla_{w'} (\nabla_{w'} \mathcal{L}_\tau(w'))\|, \\
\|w_\tau^*(w)\| &= \|w - \alpha \nabla_w \mathcal{L}_\tau(w)\| = \|w\| + \alpha \|\nabla_w \mathcal{L}_\tau(w)\|.
\end{aligned}
$$

It is easy to see that $w_\tau^*(w)$ is also Hessian-Lipschitz continuous, bounded and Lipschitz continuous.

Applying Lemma 3 in (Collins et al., 2020), we obtain that $\mathcal{L}_\tau(w_\tau^*(w))$ is $C$-smooth, where $C$ is positive. Then setting $C = L$ and using Theorem 4.2, we get the desire result.

$\square$

## D  HYPER-PARAMETER SELECTION OF SIMOL

For the few-shot regression experiment (section 5.1), the regularization parameters $\hat{\lambda}'$, $\hat{\lambda}''$ are selected from $\{0.001, 0.01, 0.1, 1\}$, and learning rate $\beta'$ is selected from $\{0.01, 0.03, 0.1, 0.3, 1\}$ based on the validation set.

For the few-shot classification experiments (section 5.2), we use the $\hat{\lambda}'$, $\hat{\lambda}''$) combination selected from few-shot regression, while the learning rate $\beta'$ is selected from $\{0.0003, 0.0008, 0.01, 0.03, 0.08, 0.1\}$ for the 1-shot *miniImageNet* task based on the validation set. this is then also used in the other few-shot classification experiments.

