# OpenReview forum: "Enhancing Meta Learning via Multi-Objective Soft Improvement Functions"
_ICLR.cc/2023/Conference — ICLR 2023 poster_

### Official Review · Reviewer_4MPs · 2022-10-24

**Confidence:** 4
**Correctness:** 3
**Technical Novelty And Significance:** 2
**Empirical Novelty And Significance:** 3
**Recommendation:** 6

**Clarity, Quality, Novelty And Reproducibility:**

**Clarity:** The paper is generally well-written and well-organized.

**Quality:** The quality of the overall paper is good. But the lack of discussion for some crucial concerns listed in the weaknesses makes the contribution unclear.

**Novelty:** The multi-objective optimization formulation is a reasonable extension from closely related work, and the novel contribution on dealing with huge tasks for meta learning is not well supported by the current analysis.

**Reproducibility:** Given the recently opposite findings on multi-objective optimization for MTL, there is a concern that the proposed method could not be robust for the meta learning problem.

**Strength And Weaknesses:**

**Strengths:**

+ This paper is generally well-organized and easy to follow.

+ To my understanding, this is the first work to consider meta learning as a multi-objective optimization problem that treats each task (the number could be huge) as a single objective to optimize. But I also have some concerns on this formulation (see weaknesses below).

+ The proposed SIMOL method is simple and easy to implement.

**Weaknesses:**

**1. Novelty**

This work proposes to formulate meta learning as a multi-objective optimization problem. However, many similar works have already been done to leverage multi-objective optimization for multi-task learning [1,2,3]. Given the close relation between meta learning and multi-task learning [4], this formulation is straightforward.

The second contribution to dealing with the huge number of tasks is indeed interesting. However, the proposed scalable optimization method directly uses the improvement function [5] with the mini-batch update, which makes it like yet another multi-objective optimization method for MTL/meta learning. I also have some concerns on the proposed method's ability to deal with the huge number of objectives (see below).

**2. Multi-Objective Optimization Formulation for Better Generalization**

It is unclear why the multi-objective optimization formulation could lead to a better generalization performance for meta learning.

*i) Why Pareto solution on the training set is good for meta learning?*

In multi-task learning, the goal is to find a solution to maximize the performance for all tasks (objectives). Since the tasks could be conflicted with each other, it is reasonable to find Pareto solution(s) for the given problem. In meta-learning, however, the goal is to improve the fast adaption performance for new tasks that are not seen for the training (optimization) process. Why could a Pareto solution on the training set be good for better generalization performance for unseen tasks?

Is it meaningful to formulate meta-learning as a multi-objective optimization problem with huge objectives, especially when the tasks we actually care about (e.g., new unseen tasks) are not in this huge set of objectives? A clear discussion on this motivation could be very helpful.

*ii) Why could the simple linear scalarization approach be worse than other methods for meta learning?*

To my understanding, the simple linear scalarization approach can still find (stationary) Pareto solutions, although it is true that it cannot find any Pareto solution on the non-convex part of the Pareto front. Since the goal here is to find a single Pareto solution (but not to find the whole Pareto front), it is unclear why the solution found by linear scalarization will perform worse than the solutions found by other multi-objective optimization methods. Different Pareto solutions should be non-dominated with each other, and it is hard to tell which one should be better than the others, especailly for the performane of unseen tasks.

*iii) Why the improvement function method could be better to handle meta learning with huge tasks?*

The improvement method used in this work is a non-scalarization multi-objective optimization algorithm, which aims to find a common descent direction for **all** objective functions. Many other multi-objective optimization algorithms also have the same goal but with different mechanisms. Given its min-max structure, the improvement method is closely relative to the Chebyshev method for multi-objective optimization [6] (now with equal preferences and adaptive reference point), which has been used for MTL [7,8]. Why could it have better performance for meta learning?

*iv) Why a single Pareto solution could be good for all tasks?*

Closely related to the previous points, since there is no single best solution for all objectives, most multi-objective optimization methods aim to find a good approximation (e.g., a set of representative solutions with different optimal trade-offs) for a given problem [6]. See the related approaches for multi-task learning [9, 10, 11] and also meta multi-objective reinforcement learning [12].

The motivation for finding only one single Pareto solution to have the best overall performance for all objectives (and even the new objectives that the algorithm has never seen) is not solid.

**3. Algorithm Performance**

The closely related work that takes meta-learning as multi-task learning [4] should be included as a baseline. Since this work is "one step further" from [4], why not directly develop the SIMOL method based on [4] which could be more efficient and straightforward than the bilevel formulation?

**Much more importantly**, due to the opposite findings in some closely related work, a solid analysis could be needed to support the proposed algorithm's promising performance. The key motivation of this work is to use a multi-objective optimization algorithm to solve the meta-learning problem, and some promising performances have been reported for different problems. However, the current work on multi-task learning leads to an opposite finding. According to [13,14,15], a simple linear scalarization method can perform comparably or even better than all those multi-objective optimization methods for both supervised learning and reinforcement learning. The effect of fine-tuned hyper-parameters (rather than the multi-objective optimization formulation) could lead to significantly different algorithm performances. Due to the close relation between multi-task learning and meta-learning, strong and solid evidence and discussion should be provided to support the finding in this work.


**4. Theory**

This work leverage the theoretical analysis from the improvement function method [5] to give the (Pareto stationary) convergence analysis, and Lin et al. [16] to give the convergence analysis for mini-batch update. However, the more important issues, such as the connection from the Pareto solution to better generalization performance for meta-learning, and the reason why the Pareto solution found by simple linear scalarization could have poorer generalization performance, are not discussed.

**Other Comments**

1. In Figure 1, if batch size =1, both mini-batch MGDA and SIMOL will only have one objective value (either f1 or f2) and its gradient at each step. There should be only a single term in (2) for SIMOL without any maximization comparison. Why did they perform quite differently in this case?

2. It is also interesting to know the performance of the best-10% for different problems.

3. It seems that a ")" is missing in the definition of \bar d^* two lines below (16).

[1] Multi-task learning as multi-objective optimization. NeurIPS 2018.

[2] Gradient surgery for multi-task learning. NeurIPS 2020.

[3] Conflict-averse gradient descent for multi-task learning. NeurIPS 2021.

[4] Bridging multi-task learning and meta-learning: Towards efficient training and effective adaptation. ICML 2021.

[5] Interactive bundle-based method for nondifferentiable multiobjeective optimization: NIMBUS. Optimization 1995.

[6] Nonlinear multiobjective optimization. Kluwer 1998.

[7] Tchebycheff Procedure for Multi-task Text Classification. ACL 2020.

[8] A Multi-objective / Multi-task Learning Framework Induced by Pareto Stationarity. ICML 2022.

[9] Pareto multi-task learning. NeurIPS 2019.

[10] Efficient Continuous Pareto Exploration in Multi-Task Learning. ICML 2020.

[11] Learning the Pareto Front with Hypernetworks. ICLR 2021.

[12] Meta-Learning for Multi-objective Reinforcement Learning. IROS 2019.

[13] A closer look at loss weighting in multi-task learning. arXiv:2111.10603, 2021.

[14] In Defense of the Unitary Scalarization for Deep Multi-Task Learning. arXiv:2201.04122, 2022.

[15] Do Current Multi-Task Optimization Methods in Deep Learning Even Help? arXiv:2209.11379, 2022.

[16] On gradient descent ascent for nonconvex-concave minimax problems. ICML 2020.


**Summary Of The Paper:**

This work proposes SIMOL, a soft improvement based multi-objective optimization algorithm, for efficient meta learning. The proposed method first formulates meta learning as a multi-objective optimization problem, where each task is an objective to optimize. Then, to handle the huge number of tasks (e.g., can be up to 7 x 10^6) in meta-learning, it further leverages the soft improvement algorithm with mini-batch update as its scalable gradient-based optimizer for model training. Theoretical analyses and experimental studies have been conducted to show the good properties and practical performance of the proposed method.


**Summary Of The Review:**

To my understanding, this is the first work to consider meta learning as a multi-objective optimization problem that treats each task (the number could be huge) as a single objective to optimize. However, due to the major concerns on the multi-objective optimization formulation, algorithm performance, and theoretical analysis, I cannot vote to accept the current manuscript.

---

> ### Author Response · Authors · 2022-11-18
> **Responses to Reviewer 4MPs (1/3)**
>
> We thank the reviewer’s helpful feedback on our submission. We summarize the reviewer’s questions and the following is our responses:
>
> ---
> ***Q1:  the proposed scalable optimization method directly uses the improvement function [5] with the mini-batch update.***
>
> There might be some misunderstanding by the reviewer. We did NOT directly use the improvement function. As mentioned at the beginning of page 5, the improvement function is defined as $\max_{\tau=1\dotsc m}\{\mathcal{L}_\tau (w_\tau^* (w))-\mathcal{L}_\tau (w_\tau^*(w'))\}$, where $m$ is the number of tasks. Hence, direct use of the improvement function requires accessing all the tasks in every epoch, which can be very expensive and even infeasible.
>
> Indeed, how to make the improvement function scalable to a large number of tasks is our main
> contribution. As described in Section 3.1, we achieve this by
> (i) relaxing the max in the definition of the improvement function to softmax via the introduction of the re-weight value $r(\tau)$ (soft improvement function);
> (ii) parameterizing $r(\tau)$ by a re-weighting network $r_\theta(\tau)$ so as to obtain every $r_\theta(\tau)$ rapidly;
> (iii) designing a regularizer and loss function to train $\theta$ to guarantee $r_\theta$ stay close to real $r$.
>
> ---
> ***Q2: Why Pareto solution on the training set is good for meta learning? ...
> In meta-learning, however, the goal is to improve the fast adaption performance for new tasks that are not seen for the training (optimization) process.
> Why could a Pareto solution on the training set be good for better generalization performance for unseen tasks?***
>
>
> While fast adaption to new tasks is important, the standard meta-learning setting requires the training and new test tasks to follow the same distributions as in:
>
> - "Theoretical Convergence of Multi-Step Model-Agnostic Meta-Learning". JMLR, 23 (2022): 29-1.
> - "Meta-learning with implicit gradients". NeurIPS-2019.
> - "Bilevel optimization: Convergence analysis and enhanced design." ICML-2021.
>
> While there are some meta-learning papers on out-of-distribution samples, we focus on the standard in-distribution setting in this paper.
>
> With the in-distribution assumption, we have the following new proposition, which is added in
> Proposition C.1 in this revised version (proof is in Appendix C).
> This indicates that $w^*$, obtained in the training set by SIMOL, can generalize to unseen testing datasets.\
> If the distribution over training tasks $P(\mathcal{T_{train}} )$ and the distribution over testing tasks $P(\mathcal{T_{test}} )$ are the same, when $E_{\tau \sim P(\mathcal{T_{train}} )}\mathcal{L_\tau} (w_\tau ^* (w^*  ))\leq \epsilon $,
> where $w^*$ is the parameter obtained by SIMOL,
> we have $E_{\tau \sim P(\mathcal{T_{test}} )}\mathcal{L_\tau  }(w_\tau ^* (w^* ))\leq \epsilon $.
>
> ---
> ***Q3: is it meaningful to formulate meta-learning as a multi-objective optimization problem  especially when the tasks we actually care about (e.g., new unseen tasks) are not in this huge set of objectives?***
>
> As mentioned above, with the in-distribution assumption, even when the new unseen task may not be one of the training tasks (objectives), it should be from the same distribution as the training tasks.
> The proposition in the reply to Q2 shows that the solution obtained by the proposed algorithm can be generalized to unseen testing datasets.
>
> ---
> ***Q4: Why could the simple linear scalarization approach be worse than other methods for meta learning?***
>
> As mentioned in our introduction and related works, linear scalarization is known to suffer from the "conflicting gradients" problem (Yu et al., 2020; Liu et al., 2021a), in that a few of the task gradients may dominate the update.
> See also (Sener & Koltun, 2018; Navon et al., 2022; Liu et al., 2021b).
> This is also empirically demonstrated in Section C.2 (Section D.2 of this revised version). As can be seen from Figure 5, linear scalarization may converge to an extreme point on a concave Pareto front (this is also mentioned in  (Emmerich & Deutz, 2018)).
> Tables 3-4 also show that
> the performance of
> linear scalarization
> on the
> worst-10\% tasks
> is worse than that of
> SIMOL, indicating that
> linear scalarization does ignore many tasks in large-scale tasks environments.

---

> > ### Author Response · Authors · 2022-11-18
> > **Responses to Reviewer 4MPs (2/3)**
> >
> > (Continued)
> >
> > ---
> > ***Q5: Why the improvement function method could be better to handle meta learning with huge tasks?
> > ... the improvement method is closely relative to the Chebyshev method for multi-objective optimization [6] (now with equal preferences and adaptive reference point), which has been used for MTL [7,8]. Why could it have better performance for meta learning?***
> >
> > First, note that we do not simply use the original improvement function, which also cannot handle a huge number of objectives. Instead, a number of modifications are made (see our reply to Q1).
> >
> >
> > The Chebyshev method [6,7,8] also cannot handle a huge number of tasks. Note that the computational complexity per epoch for [6,7,8] is $O(m^2)$, where $m$ is the number of tasks. On the other hand,  SIMOL's computational complexity per epoch is only $O(B)$, where $B$ is the batch size.
> >
> >
> > ---
> > ***Q6: Why a single Pareto solution could be good for all tasks?
> > Closely related to the previous points, since there is no single best solution for all objectives, most multi-objective optimization methods aim to find a good approximation... The motivation for finding only one single Pareto solution to have the best overall performance for all objectives (and even the new objectives that the algorithm has never seen) is not solid.***
> >
> > We agree that it is hard for a
> > single Pareto solution to be good for all tasks. Ideally, one should find the
> > whole pareto set in a multiobjective optimization problem. However, (i) finding
> > the
> > whole pareto set efficiently for a general multiobjective optimization problem is still an
> > open research issue ("Approximation methods for multiobjective optimization problems: A survey." INFORMS Journal on Computing 33.4 (2021): 1284-1299).
> > (ii)
> > Existing meta-learning algorithms (such as
> > MAML) also try to find a single solution.
> > thus, we follow the same setting.
> >
> > As discussed above,
> > the main problem with the single solution
> > obtained by linear scalarization is that it can suffer from the "conflicting
> > gradients" problem. Hence, while this paper still aims to find a single solution,
> > the goal is that this solution will not be dominated by just a few
> > objectives.
> > Empirically, for the example in Appendix D.2, Figure 5 shows that SIMOL can find
> > different Pareto points by using different random seeds, while
> > linear scalarization
> > (with different random seeds)
> > can only find extreme points.
> > Tables 1,3,4 also indicate that
> > SIMOL outperforms the "min average loss MAML" in terms of performance on the
> > worst-10\% tasks.
> >
> > ---
> > ***Q7: The closely related work that takes meta-learning as multi-task learning [4] should be included as a baseline.***
> >
> > As suggested by the reviewer, we added the method in [4] (denoted as MTL) as a baseline in Tables 3 and 4 of this
> > revised version. Results show that
> > SIMOL also outperforms MTL in terms of the overall performance
> > and worst-10\% performance.
> >
> > ---
> > ***Q8: why not directly develop the SIMOL method based on [4] which could be more efficient and straightforward than the bilevel formulation?***
> >
> > Recall that the objective for MTL in [4] is
> >
> > $\min_{w,[w_{\tau }]_{\tau=1}^{m}}\sum_\tau l_\tau( D_\tau ,w\cup w_\tau )$,
> >
> > where $w_\tau $ is the weight for the task-specific output layer.
> >  With a huge number of tasks, one cannot store all of $[w_{\tau}]_{\tau = 1}^{m}$.
> >
> > For few-shot classification tasks, they
> > represent every task
> > as a combination of a small number of classes (details in Sec 5.2 in [4]), and so
> > they only need to store the linear classifiers
> > for these classes, which are much smaller than storing  $[w_{\tau }]_{\tau = 1}^{m}$.
> > However, this cannot be used on large-scale regression and reinforcement learning
> > tasks as these tasks do not have classes, and thus the trick in [4] cannot be
> > used.
> >
> > ---
> > ***Q9: The opposite findings in some closely related work...According to [13,14,15],
> > a simple linear scalarization method can perform comparably or even better than
> > all those multi-objective optimization methods for both supervised learning and
> > reinforcement learning.***
> >
> > Experiments in [13,14,15] are only run on small environments. Specifically,
> > the largest number of tasks used in [13,14,15] are 40, 50, and 40, respectively.
> > On the other hand, we have around $7\times 10^6$ tasks in our experiments. In this
> > paper, we are interested in handling a huge number of
> > tasks.
> >
> > ---
> > ***Q10: The effect of fine-tuned hyper-parameters (rather than the
> > multi-objective optimization formulation) could lead to significantly different
> > algorithm performances.***
> >
> >
> > For the baselines,
> > their hyperparameters are selected
> > by following the corresponding original papers.
> > While further fine-tuning hyper-parameters may improve the performance of the baselines,
> > this may not be very likely as these baselines (i.e., MAML and its variants) have been popularly used
> > and their hyper-parameters have already been fine-tuned by existing works (Nichol & Schulman, 2018; Zhou et al., 2019; Zhou
> > et al., 2021; Rajeswaran et al., 2019).

---

> > > ### Author Response · Authors · 2022-11-18
> > > **Responses to Reviewer 4MPs (3/3)**
> > >
> > > (Continued)
> > >
> > > ---
> > > ***Q11: Theory. The more important issues, such as the connection from the Pareto
> > > solution to better generalization performance for meta-learning.***
> > >
> > > In this revised version, we add Prop. C.1, which shows that with the
> > > in-distribution assumption, the solution obtained by the proposed algorithm can generalize to unseen testing datasets. Please refer to our response to Q2 for more details.
> > >
> > >
> > > ---
> > > ***Q12: In Figure 1, if batch size =1, both mini-batch MGDA and SIMOL will only have one objective value (either f1 or f2) and its gradient at each step. There should be only a single term in (2) for SIMOL without any maximization comparison. Why did they perform quite differently in this case?***
> > >
> > > If batch size =1,
> > > minibatch-MGDA degenerates to $w^{s+1} = w^{s} -\beta \nabla_{w}\mathcal{L_{\tau_i}}(w_{\tau_i}^* (w^s)) $, where $\tau_i$ is the
> > > objective used in epoch $i$.  On the other hand, from
> > > Alg. 1,
> > > the update of SIMOL is
> > > $w^{s+1} =w^s+\beta \frac{1}{B} d^* =  w^{s} -\beta r_\theta(\tau_i) \nabla_{w^s}\mathcal{L_{\tau_i}}(w_{\tau_i}^{*} (w^s)) $.
> > > Hence,
> > > the SIMOL update differs from
> > > minibatch-MGDA update by having
> > > a dynamic weight $r_\theta(\tau_i)$.
> > >
> > > ---
> > > ***Q13: It is also interesting to know the performance of the best-10\% for different problems.***
> > >
> > > As suggested, we added the performance of the best-10$\%$ on the few-shot
> > > classification experiment in Tables 7-8. As can be
> > > seen, SIMOL still outperforms the other baselines. However, note that the improvement here is smaller than that for the worse-10\%, indicating a larger improvement on tasks with poor performance.
> > >
> > > ---
> > > ***Q14: It seems that a ")" is missing in the definition of $\bar d^*$ two lines below (16).***
> > >
> > > Thank you for pointing it out, and we have fixed it in the updated version.

---

> ### Comment · Reviewer_4MPs · 2022-11-22
> **Follow-up Comments**
>
>
> Thank you for your thorough response and new experimental results. Some of my concerns have been appropriately addressed, so I raise my score to 5.
>
> Here are the follow-up comments on the remaining concerns.
>
> **Q1&5 SIMOL v.s. Original Improvement Function**
>
> It is crucial to show the performance of the original improvement function with the simple mini-batch update (similar to the mini-batch linear scalarization, and now only takes the max for the sampled subset of tasks at each step). This result could strongly support the main contribution of the proposed SIMOL method.
>
> An ablation study on the different components (e.g., soft improvement function, re-weighting network, regularizer, and loss function) could be also useful.
>
> **Q2&3 Why Pareto Solution for Meta-Learning**
>
> The concern here is that a Pareto solution (found by SIMOL) is not necessarily the solution that minimizes $E(\mathcal{L})$ for all task loss $\mathcal{L}$, even on the distribution over training tasks.
>
> In addition, with the strong in-distribution assumption, I think it is not needed to write the straightforward conclusion into a formal Proposition C.1 and proof.
>
> **Q4&6&9 Why Not Simple Linear Scalarization**
>
> Is there any theoretical analysis to show that the solution found by SIMOL can achieve better  $E(\mathcal{L})$ than the solution found by linear scalarization?
>
> As in my original comments, I agree that the linear scalarization cannot find any Pareto solution on the non-convex part of the Pareto front, but its solution is still a Pareto solution. Therefore, it is unclear why some Pareto solutions (e.g., found by SIMOL) could be better than others (e.g., found by linear scalarization) when they are indeed nondominated by each other.
>
> The current findings in [13,14,15] show that a simple linear scalarization method can perform comparably or even better than all those multi-objective optimization methods, and indicate that the "conflicting gradients" is indeed not an issue for linear scalarization's performance. Why could it be different for the meta-learning setting (with a huge number of tasks)?
>
> **Q12 Experiment in Figure 1**
>
> It is still a bit unclear how the dynamic weight in SIMOL can avoid conflicting gradients at each step with mini-batch 1 and lead to a smooth convergence trend.

---

> > ### Author Response · Authors · 2022-11-25
> > **Responses to the Follow-up Comments (1/2)**
> >
> > Thank you for increasing the score! We also appreciate your follow-up comments to help further improve our paper. Here is our response to your questions:
> >
> >
> > ---
> > ***Q15: It is crucial to show the performance of the original improvement function with the simple mini-batch update (similar to the mini-batch linear scalarization, and now only takes the max for the sampled subset of tasks at each step).***
> >
> > We implement the mini-batch improvement function in the few shot regression tasks:
> >
> >
> > ***Table A: MSE  (with 95\% confidence interval)  for few-shot regression.***
> > |                   |       overall         |                 |                              worst-10\%            |                       |
> > |-------------------|-----------------|-----------------|-----------------------|-----------------------|
> > |                   | 5-shot          | 2-shot          | 5-shot                | 2-shot                |
> > | Mini-batch improvement function | $0.69 \pm 0.03$  | $1.79 \pm 0.09$  |  $2.63 \pm 0.21$ | $8.23 \pm 0.36$ |
> > | SIMOL | $0.34\pm 0.04$  | $1.24 \pm 0.08$  |  $1.69 \pm 0.18$ | $5.66 \pm 0.33$ |
> > | min average loss | $ 0.43 \pm 0.11$  | $ 1.70 \pm 0.11$  |  $ 2.13 \pm 0.21$ | $ 7.75 \pm 0.48$ |
> >
> >
> >
> > ***Table B: AUC  (with 95\% confidence interval)  for few-shot classification in miniImageNet.***
> > |                   |       overall         |                 |                              worst-10\%            |                       |
> > |-------------------|-----------------|-----------------|-----------------------|-----------------------|
> > |                   | 5-shot          | 1-shot          | 5-shot                | 1-shot                |
> > | Mini-batch improvement function | $27.71\pm 0.40$ | $23.28 \pm 1.00$  |  $13.67 \pm 0.61$ | $10.20 \pm 0.98$ |
> > | SIMOL |  $65.83 \pm 0.86$  | $50.62\pm 1.39$ |  $44.81\pm 0.58$ | $14.99 \pm 1.72$ |
> > | min average loss | $62.13 \pm 0.72$  | $ 49.24 \pm 0.78$  |  $ 41.71 \pm 1.02$ | $13.33 \pm 1.07$ |
> >
> > As can be seen,
> > on few-shot regression,
> > both the overall and worst-10\% MSEs are higher than the proposed
> > SIMOL.
> > The overall and worst-10\% AUC are lower than ours and "min average loss".
> > This is because for the standard improvement function, replacing the
> > "max over all tasks" by "max over a mini-batch of tasks" can  have very different solutions, and thus
> > worse performance.
> > On the other hand,
> > for the proposed method,
> > Proposition 3.3 shows that the quality of the mini-batch approximation can be
> > bounded.
> >
> > We will update our results to the main text in the revised version.
> >
> >
> >
> >
> > ---
> > ***Q16: An ablation study on the different components (e.g., soft improvement function, re-weighting network, regularizer, and loss function) could be also useful.***
> >
> >
> > As suggested, we add the following ablations:
> >
> > 1) Ablation on improvement function: please see the reply to Q15 on the comparison between the standard improvement function and the proposed soft improvement function.
> >
> > 2) Ablation on  re-weighting network  structure:
> > Appendix D.3 in the original submission contains an ablation on the number of layers in the re-weighting network for few-shot regression.
> > Results show that using a simple 2-layer network is good enough.
> >
> >
> >
> > 3)
> > Ablation on the regularizer: we compare the performance of SIMOL with and without
> > the "regularizer" (i.e., the term $\frac{\hat{\lambda}^{\prime \prime}}{2}(\frac{1}{|B|} \sum_{\tau \in B} r_\theta(\tau)-1)^2$ in $\tilde{K}_B(\theta)$).  However,
> > without this term,
> > SIMOL  does not
> > converge.
> > The reason is that as shown in section 3.1, we introduce this term NOT as a regularizer, but as a soft
> > constraint for the constraint that  $\sum_\tau r_\theta(\tau)=1$. Without
> > this (soft) constraint, the descent direction can be incorrect.
> >
> >
> >
> > ---
> > ***Q17: The concern here is that a Pareto solution (found by SIMOL) is not necessarily the solution that minimizes  for all task loss, even on the distribution over training tasks.***
> >
> > We agree that the Pareto solution found by SIMOL may not
> > minimize the total loss over all tasks. However, recall that our goal is to downweight the
> > effect of a few extreme
> > dominating tasks. Indeed, this is similar to computing the mean of a set of numbers. If
> > some numbers in the set are extremely big (outliers), the mean (which also minimizes the average
> > distance to all the numbers) will be dominated by those extreme numbers. This is
> > often not desirable. Oftentimes, a trimmed mean is more desirable than the mean.
> > Hence, when one minimizes the total loss over all tasks, the solution can be
> > problematic when there are outlying tasks.
> >
> > ---
> > ***Q18: In addition, with the strong in-distribution assumption, I think it is not needed to write the straightforward conclusion into a formal Proposition C.1 and proof.***
> >
> > We will remove Proposition C.1 in the revised version.

---

> > > ### Author Response · Authors · 2022-11-25
> > > **Responses to the Follow-up Comments (2/2)**
> > >
> > > (Continued)
> > >
> > > ---
> > > ***Q19: Is there any theoretical analysis to show that the solution found by SIMOL can achieve better than the solution found by linear scalarization?***
> > >
> > > To our best knowledge,
> > > none of the related methods, such as (Yu et al., 2020; Liu et al., 2021a, Navon et al., 2022; Liu et al., 2021b),
> > > provide
> > > theoretical results on why they are better than linear scalarization. We think that would be an important future work.
> > >
> > > ---
> > > ***Q20: As in my original comments, I agree that the linear scalarization cannot find any Pareto solution on the non-convex part of the Pareto front, but its solution is still a Pareto solution. Therefore, it is unclear why some Pareto solutions (e.g., found by SIMOL) could be better than others (e.g., found by linear scalarization) when they are indeed nondominated by each other.***
> > >
> > > We certainly agree that by definition, one Pareto solution cannot dominate another Pareto
> > > solution, and so no solution is universally better than others. However, what we
> > > are studying in this paper is that there might be a few tasks that can contribute
> > > significantly to the solution (just like a few outliers can
> > > contribute significantly to the computation of the mean. Please see our reply to
> > > Q17). As discussed in the
> > > related works (such as (Yu et al., 2020; Liu et al., 2021a)), this is the problem
> > > suffered by linear scalarization, and this is the problem that we want to address
> > > in this paper. As
> > > demonstrated in Figure 5,
> > > linear scalarization (even with different random seeds) always find extreme
> > > points.
> > >
> > >
> > > ---
> > > ***Q21: The current findings in [13,14,15] show that a simple linear scalarization method can perform comparably or even better than all those multi-objective optimization methods, and indicate that the "conflicting gradients" is indeed not an issue for linear scalarization's performance. Why could it be different for the meta-learning setting (with a huge number of tasks)?***
> > >
> > > That depends on the dataset. Just like the example mentioned in the reply to Q17
> > > above, the simple mean can often be a good average for a set of numbers. However, in the
> > > presence of outliers, the mean can be dominated by the outliers.
> > > Hence, it is possible that linear scalarization works well in some problems.
> > > However, as demonstrated in our experimental results in section 5, linear
> > > scalarization (min average loss) has inferior performance.
> > >
> > >
> > > ---
> > > ***Q22: It is still a bit unclear how the dynamic weight in SIMOL can avoid conflicting gradients at each step with mini-batch 1 and lead to a smooth convergence trend.***
> > >
> > >
> > > The dynamic weight in SIMOL comes from our theoretical analysis. Hence, its
> > > usefulness
> > > is due to the theoretical results in
> > > Sec. 4.

---

> > > > ### Comment · Reviewer_4MPs · 2022-11-25
> > > > **Thank you for your thorough response**
> > > >
> > > > Thank you for your thorough response. All of my concerns have been properly addressed so I raise my score to 6. The connection between the Pareto solution and better generalization performance could be important for future work.

---

> > > > > ### Author Response · Authors · 2022-11-25
> > > > > **Thank you**
> > > > >
> > > > > We are glad to hear that our responses addressed your concerns. Thank you again for raising the score!

---

### Official Review · Reviewer_f9rZ · 2022-10-26

**Confidence:** 3
**Correctness:** 3
**Technical Novelty And Significance:** 3
**Empirical Novelty And Significance:** 3
**Recommendation:** 8

**Clarity, Quality, Novelty And Reproducibility:**

# Clarity
The paper is written pretty clearly. The text is not difficult to understand,

# Quality
The algorithm seems mathematically sound, although I did not closely check the proofs. The experiments are fairly complete in terms of evaluation metrics and diversity of environments, although no ablations are included. SIMOL provides improvements that are often statistically significant.

# Novelty
I am not completely up-to-date with the literature on multi-task optimization, but as far as I know the use of improvement functions for meta-learning is novel and presents a new class of algorithms.

# Reproducibility
The authors do not provide code, but provide pseudocode and hyperparameters. I don't think reproducing the results would be an issue.

**Strength And Weaknesses:**

# Strengths
- The worst task performance of SIMOL improves over those of vanilla MAML and Prototypical Networks without increasing the computational burden drastically. The overall performance with a PN backbone is also better.
- The algorithm is well mathematically motivated, and theoretical analysis of convergence is provided along with experiments.
- The paper is well written and the arguments are straightforward to understand.

# Weaknesses
- There are no ablation studies, e.g. on the architecture of the network
- For the MAML backbone, the improvement of SIMOL is often not statistically significant
- The convergence theory may not be tight; the rate is slower than indicated from experiments

# Questions
- Are gradient signs missing from line 5 of Alg. 1?
- How are the hyperparameters chosen?
- Why is PCGard not included as a baseline?


**Summary Of The Paper:**

This paper considers solving meta-learning (formulated as bilevel optimization with the inner level being tasks) using techniques from multi-objective optimization (MOO). Previous work along this line required computing gradients for each task at each iteration, which is expensive. This paper proposes an algorithm to find an $\epsilon$-Pareto stationary point using only mini-batches of tasks, built on the concept of improvement functions from MOO. Experiments are done on linear regression, few-shot classification, and meta-reinforcement learning. The proposed algorithm SIMOL generally improves over MAML and Prototypical Networks without significantly increasing computational burden, unlike previous work.

**Summary Of The Review:**

This paper is a novel application of approaches from MOO to meta-learning, with the intuition that a Pareto optimality would also lead to faster adaptation at test time. The algorithm is mathematically motivated and analyzed, and shown to improve upon MAML with similar computational complexity. However, SIMOL does not always lead to statistically significant improvements and no ablation studies are carried out.

---

> ### Author Response · Authors · 2022-11-18
> **Responses to Reviewer f9rZ**
>
> Thank you for your constructive and positive reviews. We summarize the reviewer's questions and present our responses below.
>
> ---
> ***Q1: There are no ablation studies, e.g., on the architecture of the network.***
>
> Regarding ablations, a study on the effect of batch size is shown in Table 5 of the original submission (Table 6 in this updated version).
>
> As suggested by the reviewer, we added an ablation study on the number of layers in the re-weighting network. The overall MSE and worst-10\% MSE are shown in Table 5 of this updated version. As can be seen, using a simple 2-layer network is good enough.
>
> ---
> ***Q2: For the MAML backbone, the improvement of SIMOL is often not statistically significant.***
>
> There was a typo in Table 3. The 5-shot standard deviation of SIMOL should be 0.86 (instead of 1.86 in the earlier version), and is comparable with the other baselines.
>
> Moreover, as suggested, we add statistical significance test (matched-pair t-test) for the few-shot classification results in Tables 3 and 4. SIMOL consistently outperforms all the baselines at 0.1 significance level.
>
> ---
> ***Q3: The convergence theory may not be tight; the rate is slower than indicated from experiments.***
>
> We agree that the upper bound may not be tight. However, the bound requires minimax analysis, and our bound is similar to those in recent minimax analysis papers (Theorem 4.9, "On gradient descent ascent for nonconvex-concave minimax problems", ICML-2020).
>
> ---
> ***Q4: Are gradient signs missing from line 5 of Alg. 1?***
>
> Thank you for your careful reading. A sign is missing, and we have added it back in this revised version.
>
> ---
> ***Q5: How are the hyperparameters chosen?***
>
> Hyperparameters for the backbone meta-learning algorithms (MAML and prototypical network) follow the original papers (Finn et al., 2017; Snell et al., 2017). This is now clarified in Sections 5.1 and 5.2 of this revised version.
>
> As for the re-weighting network, on the 2-shot regression tasks (Section 5.1), $\hat{\lambda}' $ and
> $\hat{\lambda}''$ are selected from $\{0.001,0.01,0.1,1\}$, and the learning rate $\beta'$  is selected from $\{0.01,0.03,0.1,0.3,1\}$. We then directly use the best $\hat{\lambda}', \hat{\lambda}$ values obtained for the 5-shot regression task in Section 5.1 and other classification tasks in Section 5.2.
>
>
>
> Moreover, for the classification experiment, the learning rate $\beta'$ is selected from $\{0.0003,0.0008,0.01,0.03,0.08,0.1\}$ in the 1-shot miniImageNet classification experiment in Section 5.2. The best $\beta'$ value obtained is then used for the other classification experiments in Section 5.2.
>
> These details are now added to Appendix D.1 in the updated version.
>
> ---
> ***Q6: Why is PCGard not included as a baseline?***
>
>
> Liu et al. (2021) show that PCGrad does not perform better than CAGrad, and thus was not included in the original submission.
>
> As suggested by the reviewer, we now add PCGrad as an additional baseline in the few-shot regression
> experiments in Section D.3.
> Results are shown in Table 9. As can be seen, PCGrad performs similarly to CAGrad and is outperformed by the proposed method.

---

> > ### Comment · Reviewer_f9rZ · 2022-11-19
> > **Thank you to the authors for the response**
> >
> > I feel that the authors adequately answered my questions and am happy to keep my score. In addition, I agree with reviewer 4MPs that the main paper should discuss why Pareto optimality is helpful for meta-learning, as it is a primary motivation of the algorithm.

---

> > > ### Author Response · Authors · 2022-11-20
> > > **Thank you**
> > >
> > > Thank you again for the positive review and feedback.
> > >
> > > Regarding why Pareto optimality is helpful for meta-learning, as meta-learning involves a number of tasks, the goal should be to obtain a Pareto (non-dominated) solution (i.e., the loss on each task is no worse than that of any other solution, and the loss on at least one task is better). In multi-objective optimization terminology, the meta-learning solution should be Pareto-optimal. However, even when a solution is Pareto-optimal, it may be mainly dominated by a few of the objectives (tasks). This is the so-called "conflicting gradients" problem (Yu et al., 2020; Liu et al., 2021a) suffered by the linear scalarization. In this paper, we are concerned with how to find a Pareto solution that is NOT mainly dominated by a few tasks.
> > >
> > > More details are in our responses to Q2-Q4 for Reviewer 4MPs.

---

### Official Review · Reviewer_sFDS · 2022-11-03

**Confidence:** 2
**Correctness:** 3
**Technical Novelty And Significance:** 3
**Empirical Novelty And Significance:** 3
**Recommendation:** 6

**Clarity, Quality, Novelty And Reproducibility:**

The overall idea is clearly conveyed, though the massive equations hurts the readability a bit.

The idea of converting the mete-learning problem as a multi-objective optimization sounds new, and the derived algorithm with theoretical guarantee is a solid contribution too.

Algorithm 1 seems easy to reimplement, so the results could be reproducible with some effort.

**Strength And Weaknesses:**

Strength:

1. This work is well motivated that reformulating the original problem to a MOO can avoid the compromising phenomenon.
2. The overall logic of deriving the solution along the journey is quite clear: 1) The improvement function can provide a closed-form gradient for the original problem; 2) The closed-form requires  uniform sampling of the tasks, which can be parameterized as a neural network; 3) Optimizing the network requires another sampling of tasks to compute the gradient of the NN; 4) Theoretically, the gap between the full batch and mini-batch of tasks can be bounded, so we can safely adopt the SGD to optimize the NN and finally solve the MOO problem with pareto optimality.
3. Empirical studies looks very promising that the proposed method works quite well on multiple tasks and is better than the competitors.


Weaknesses:

1. There are tons of math equations. The author may need to simplify or condense them a bit, and give timely interpretation for them. It is important to clearly extract and convey the message behind the equations to the readers.
2. The fonts of Figure 1 and 2 are quite small that it is hard to read.
3. Typos: e.g., “the converged solution may be comprised” —> “the converged solution may be compromised”
4. Immediately after Equation (7), it is logically not correct to say “We first rewrite the above as:”, since Lemma 3.1 is to prove that (7) and (8) are equivalent, but before the proof, we don’t know if that’s true.
5. Definition 2.1. and 2.2. may need to cite some classical literatures rather than those new ones, as these definitions should exist long time ago.
6. In Table 3, SIMOL has much large variance than others in the MAML category. Would that be a concern?

**Summary Of The Paper:**

This paper proposes to formulate the meta-learning as a multi-objective optimization (MOO) problem instead of the traditional single objective problem where the sum of each task’s loss is optimized. By doing do the compromising phenomenon can be avoided, but it also incurs a new problem — the current gradient-based MOO solvers cannot scale to a large number of objectives. Then the authors proposes a scalable gradient-based solver via the use of improvement functions. Besides, this algorithm is guaranteed to have theoretical convergence. Experimental results on few-shot regression, few-shot classification and reinforcement learning show that the proposed method, called SIMOL, outperforms several existing algorithms in terms of either efficiency or generalization.

**Summary Of The Review:**

The paper seems to be the first to reformulate the meta-learning problem as a MOO to avoid the compromising problem. Then a gradient-based algorithm guaranteed to converge to pareto optimal solution is proposed. Overall, this is a solid contribution.

---

> ### Author Response · Authors · 2022-11-18
> **Responses to Reviewer sFDS**
>
> We much appreciate your positive and insightful comments on our paper. Here is our response to your questions point by point:
>
> ---
>
> ***Q1: There are tons of math equations. The author may need to simplify or condense them a bit, and give timely interpretation for them. It is important to clearly extract and convey the message behind the equations to the readers.***
>
> As suggested by the reviewer, we added more explanations to the theoretical results in this revised version.
>
> ---
> ***Q2: "Fonts of Figure 1 and 2 are quite small", "Typos", Definition 2.1. and 2.2. may need to cite some classical literatures rather than those new ones".***
>
> As suggested, we replaced Figures 1 and 2 with enlarged ones to make them easy to read, fixed the typo, and used an older reference.
>
> ---
> ***Q3: Immediately after Equation (7), it is logically not correct to say “We first rewrite the above as:”, since Lemma 3.1 is to prove that (7) and (8) are equivalent, but before the proof, we don’t know if that’s true.***
>
> This sentence is rewritten in this revised version.
>
> ---
> ***Q4: In Table 3, SIMOL has much large variance than others in the MAML category. Would that be a concern?***
>
> There was a typo in Table 3. The 5-shot standard deviation of SIMOL should be 0.86 (instead of 1.86 in the earlier version), and is comparable with the other baselines. Moreover, we add statistical
> significance test (matched-pair t-test) for the few-shot classification results in Tables 3 and 4. SIMOL consistently outperforms all the baselines at 0.1 significance level.

---

> > ### Comment · Reviewer_sFDS · 2022-12-01
> > **Thanks for your response.**
> >
> > The authors have addressed my main concerns, so I will keep my original rating to champion the acceptance.

---

### Author Response · Authors · 2022-11-18
**Responses to all reviewers**

We first thank so much all the reviewers for their valuable, constructive, and perspective comments. We have updated a revision of our paper, and here is a summary of our updates:

1) We added extra experiments regarding i) statistical significance testing in Tables. 3 and 4; ii) ablations study regarding different layers in Appendix D.3; iii) PCGrad in few-shot regression in Tab. 9; iv) Best-10\% in few-shot classification tasks in Tables 7 and 8.


2) We added more details regarding i) more explanations of some theoretical results (Corollary 3.3.1 and Corollary 4.2.1 ); ii) how to choose the hyper-parameters in Appendix D.1.

We sincerely hope the responses could address the comments of all the reviewers.

More discussions and suggestions on further improving the paper are also always welcomed!

Thanks again!

---

### Decision · Program_Chairs · 2023-01-20

**Decision:**

Accept: poster

**Justification For Why Not Higher Score:**

While this paper was above borderline, two reviewers thought it was a marginal accept (although one did say they would champion acceptance).  The work seems broadly interesting, but the reviewers did raise some questions including regarding the statistical significance of the results and ablation studies.

**Justification For Why Not Lower Score:**

Meta learning is broadly interesting to the community, and to that sub-field and this is a novel approach that seems to improve things.

**Metareview: Summary, Strengths And Weaknesses:**

This paper develops a new method, called SIMOL, for meta learning through using multi-objective optimization.  The authors argue that optimizing the average loss across tasks can cause the meta learning method to focus on a small set of tasks.  Thus instead they propose treating each task as a separate objective in a multi-objective optimization framework.  On a variety of experiments the authors demonstrate that SIMOL outperforms at least in the worst case on the popular MAML and Prototypical network methods.

The reviewers found the work clearly written, well motivated and technically sound.  One reviewer initially raised a number of questions, including why the multi-task optimization approach would necessarily lead to better generalization performance (as opposed to training set loss).  It appears that the authors convincingly addressed these concerns and the reviewer raised their score to an accept accordingly.  Another reviewer indicated they were willing to "champion the acceptance" after reading the author feedback.  Other weaknesses identified by the reviewers included questions about statistical significance regarding a subset of the experiments comparing to MAML and a lack of ablation studies.

Overall the reviewers were in agreement on acceptance (8, 6, 6).  Overall, this work seems broadly interesting to the community in meta learning and the fact that it's a novel approach that should be applicable on top of other approaches could make it quite useful.
 Therefore the recommendation is to accept the paper.  Hopefully the feedback from the reviewers and the responses from the discussion period will be useful in further refining the manuscript before publication.

**Note From Pc:**

if the above contains the word "oral" or "spotlight" please see: "oral" presentation means -> notable-top-5% and "spotlight" means -> notable-top-25%. As stated in our emails, we are disassociating presentation type from AC recommendations